# Investigating Neolithic caprine husbandry in the Central Pyrenees: Insights from a multi-proxy study at Els Trocs cave (Bisaurri, Spain)

Cristina Tejedor-Rodríguez[1], Marta Moreno-García[2]*, Carlos Tornero[3,4], Alizé Hoffmann[5], Íñigo García-Martínez de Lagrán[6], Héctor Arcusa-Magallón[7], Rafael Garrido-Pena[8], José Ignacio Royo-Guillén[9], Sonia Díaz-Navarro[6], Leonor Peña-Chocarro[2], Kurt. W. Alt[10,11], Manuel Rojo-Guerra[6]

**1** Instituto de Ciencias del Patrimonio (Incipit), Consejo Superior de Investigaciones Científicas (CSIC), Santiago de Compostela, Spain, **2** Instituto de Historia, Consejo Superior de Investigaciones Científicas (CSIC), Madrid, Spain, **3** Institut Català de Paleoecologia Humana i Evolució Social (IPHES), Tarragona, Spain, **4** Àrea de Prehistòria, Universitat Rovira i Virgili (URV), Tarragona, Spain, **5** Université Toulouse-Jean Jaurés, Toulouse, France, **6** Departamento de Prehistoria y Arqueología, Facultad de Filosofía y Letras, Universidad de Valladolid, Valladolid, Spain, **7** Private Technical Archaeologist, La Muela, Zaragoza, Spain, **8** Departamento de Prehistoria y Arqueología, Facultad de Filosofía y Letras, Universidad Autónoma de Madrid, Madrid, Spain, **9** Gobierno de Aragón, Zaragoza, Spain, **10** Center of Natural and Cultural Human History, Danube Private University, Krems, Austria, **11** Integrative Prehistory and Archaeological Science, University of Basel, Basel, Switzerland

\* marta.moreno@cchs.csic.es

## Abstract

Sheep remains constitute the main archaeozoological evidence for the presence of Early Neolithic human groups in the highlands of the Southern Pyrenees but understanding the role of herding activities in the Neolithisation process of this mountain ecosystem calls for the analysis of large and well-dated faunal assemblages. Cova de Els Trocs (Bisaurri, Huesca, Spain), a cave located at 1564 m a.s.l on the southern slopes of the Central Pyrenees, is an excellent case study since it was seasonally occupied throughout the Neolithic (ca. 5312–2913 cal. BC) and more than 4000 caprine remains were recovered inside. The multi-proxy analytical approach here presented has allowed us to offer new data elaborating on vertical mobility practices and herd management dynamics as has not been attempted up until now within Neolithic high-mountain sites in the Iberian Peninsula. For the first time, $\delta^{18}O$ and $\delta^{13}C$ stable isotope analyses offer direct evidence on both the regular practice of altitudinal movements of sheep flocks and the extended breeding season of sheep. Autumn births are recorded from the second half of the fifth millennium cal. BC onwards. Age-at-death distributions illustrate the progressive decline in caprine perinatal mortality together with the rising survival rate of individuals older than six months of age and the larger frequency of adults. This trend alongside the 'off-season' lambing signal at the implementation of husbandry techniques over time, probably aiming to increase the size of the flocks and their productivity. Palaeoparasitological analyses of sediment samples document also the growing reliance on herding activities of the human groups visiting the Els Trocs cave throughout the Neolithic sequence. In sum, our work provides substantial arguments to conclude that the advanced herding management skills of the Early Neolithic communities

**Data Availability Statement:** All relevant data are within the paper and its Supporting Information files.

**Funding:** This paper was supported by several projects awarded to MR-G: 'Los Caminos del Neolítico' (HAR2009-09027) and 'Los Caminos del Neolítico II' (HAR2013-46800 P) were granted by the National R&D&I Plan/Ministry of Economy, Industry and Competitiveness (https://www.ciencia.gob.es/portal/site/MICINN/) and co-financed by the Government of Aragón (https://www.aragon.es/), and 'La Memoria del Camino: Ciencia y divulgación de las primeras rutas pecuarias neolíticas en el Pirineo-MEDELCA' (FCT-2015-9947) was financed by the Spanish Foundation for Science and Technology (FECYT) (https://www.fecyt.es/). The archaeozoological research was carried out within the framework of the project awarded to MM-G '. . . y la oveja domesticó al pastor: Señales genómicas y arqueozoológicas de los primeros ovinos durante la neolitización de la península ibérica' (HAR2016-75914-R) by the National R&D&I Plan/Ministry of Economy, Industry and Competitiveness (https://www.ciencia.gob.es/portal/site/MICINN/). AH was awarded a PhD grant by the Université Fédérale de Toulouse ANR-11-IDEX-0002-02 (https://www.univ-toulouse.fr/). Support of the publication fee was granted by the CSIC Open Access Publication Initiative through its Unit of Information Resources for Research (URICI). The funders had no role in study design, data collection and analysis, decision to publish, or preparation of the manuscript.

**Competing interests:** The authors have declared that no competing interests exist

arriving in Iberia facilitated the anthropisation process of the subalpine areas of the Central Pyrenees.

## Introduction

The history of animal husbandry in the Iberian Peninsula began as early as the sixth millennium cal. BC, when farmers with domestic livestock, specifically sheep and goats, made their appearance along the Mediterranean coastal regions to rapidly reach the western and southern Atlantic shores within the 5600–5400 cal. BC interval [1–5]. Through the river valleys they also advanced fast to inland areas such as the Upper Ebro Valley and the pre-Pyrenean territories [6–13]. Increasing fieldwork and archaeological studies show the great variability of environmental and socio-economic scenarios where early Neolithic communities of different identity and origin unfolded their farming practices in Iberia [14–18]. Mountain environments are of particular interest because human impact has contributed to gradually shape these natural ecosystems into cultural landscapes [19–22]. In the case of the southern slopes of the Pyrenees, archaeological evidence of the arrival of Neolithic groups comes mainly from rock-shelters and caves, featuring tools made of polished stone, pottery vessels, cereal grains and faunal assemblages dominated by the remains of domestic animals, particularly caprines [11, 23–27] among which sheep stand out. The overwhelming abundance of sheep emphasizes not only the great adaptive capacity of the new species to upland territories but also the key part shepherding activities played in both the spread of farming and the inception of production economies in these areas [28].

One of the core issues presently under debate is whether the neolithisation process of the Pyrenean highlands was associated with the short-distance movement of livestock from sedentary mixed farming communities based in this mountain range [11] or the seasonal coming of pastoral groups from the Pre-Pyrenean lower lands and the Upper Ebro Valley [12, 24, 29, 30]. Although these hypotheses are not mutually exclusive, they correspond to divergent subsistence models according to which animal husbandry would have been practiced at very different scales (*i.e.*, the house-hold level or an intensive and more specialised level). Recognizing the full array of potential herding patterns carried out from the early Neolithic onwards and how they relate to the exploitation and occupation of the uplands requires the analysis of meaningful faunal samples from sites at high altitude. However, the available data to date on breeding practices, size of herds, culling profiles (from which management and production strategies can be understood) and on-site evidence for the mobility of flocks to the Pyrenean highlands derive from a very limited zooarchaeological record. As a result, the present state of research, primarily based on the taxonomic composition of small faunal assemblages (NISP ≤500), hampers the characterization of the full range of husbandry practices, involving multiple scales of mobility, that might have occurred soon after the advent of the first Neolithic populations [11].

Els Trocs cave, situated at more than 1500 meters above the sea level on the Central Iberian Pyrenees, is currently the only high-mountain site with a fully documented Neolithic stratigraphic sequence that has provided a large and well preserved faunal assemblage, dominated by sheep remains [23, 24]. The multi-proxy methodological approach presented in this paper focuses on the zooarchaeological and isotopic analyses of caprine remains and the palaeoparasitological analysis of sediment samples with the aim to gain new insights into the pastoral strategies carried out by the human communities who explored this ecosystem from the sixth millennium cal. BC onwards. More specifically, our goals are threefold: firstly, to provide on-

site and direct evidence for early Neolithic vertical mobility of flocks; secondly, to investigate herd management and production strategies; and thirdly, to explore the role played by mobile herding in the development of a specialized economic activity among early farming populations with agropastoral lifeways in the Pre-Pyrenean territories. Our research strategy intends that each one of the analytical proxies followed would offer specific answers to the foregoing aims and that our results will partly fill the gaps pointed above. By doing so we hope to show that the quick spread of a varied and complex production economy with well-developed agriculture and animal management techniques, bolstered the emergence of human groups ready to set up the exploitation, and perhaps even the territorial control of new settings such as the Pyrenean uplands.

## Site description, chronology and findings

Els Trocs cave is located in San Feliú de Veri (Bisaurri, Huesca), in the southern Axial Pyrenees of Aragón, Spain, where mountain peaks rise over 3000 metres above sea level (*i.e.*, Aneto 3404 m, Posets 3369 m, Maladeta 3350 m or Perdiguero 3322 m). The site stands on a conical elevation on the southern slope of Mount Els Trocs (UTM coordinates -Datum 31-: x-298.198, y-4.702.955, z-1564), in the middle of a high plateau (average altitude of 1500 m a.s.l.) with the massif of the Turbón (2492 m) to the south. This plateau is a natural perpendicular and equidistant corridor to the headwaters of the Ésera and Isábena rivers (Fig 1). The adjacent surroundings are the flatlands known as *Partida de la Selvaplana*, a territory that was traditionally used as cultivated and pastureland up to the middle of the last century.

The cave consists of a single chamber of about 15 m by 6 m with two shallow left and right secondary branches. Access is made through a steep 10 m long slope (Fig 1). Seven excavation seasons (2009–2012, 2014, 2016 and 2019) under the direction of Manuel Rojo-Guerra (Valladolid University) and José I. Royo-Guillén (Aragón Government) have uncovered a total area of about 45 m$^2$. The archaeological works have documented a singular stratigraphic sequence with four occupation phases designated as Trocs I-IV from bottom to top [23, 24, 29]. Trocs I to Trocs III date to the Neolithic period, spanning from the late sixth millennium cal. BC to the end of the fourth millennium cal. BC, while Trocs IV comprises the very scarce Roman and modern materials of the uppermost archaeological layer.

Thanks to a rigorous dating program and the analysis of the stratigraphy, we have a well-dated and established chronology for the Neolithic sequence (Fig 2 and S1 Table).

The main archaeological findings present in the 20 cm thickness of layer Trocs I are several pits of different sizes (among which two, big and round, stand out in the centre of the cave) and a floor intentionally fashioned from over 17000 plain and decorated potsherds fragments. The floor paves the entire cave and spreads over the ashes derived from different combustion events originally placed on the orange clay sediment. In addition to the large quantities of faunal remains recovered from this occupation layer, human skeletal elements of at least nine individuals (five adults, four children), all of whom show traces of peri- and post-mortem violence, were found in secondary position [23, 31] (Fig 3).

Phytolith analyses have documented a consistent use of local grass resources that were spread over the potsherds pavement with the aim of probably achieving a more comfortable surface and also, insulating it from humidity. The absence of inflorescence morphotypes from grasses suggested the collection and use of these plants before the inflorescence cells were silicified [32]. This full silicification happens at the end of the plant lifecycle, when the inflorescences are mature, which is mid to end of summer at this altitude. This data provided a clear marker for the seasonal occupation of the cave during late spring and/or beginning of summer [33].

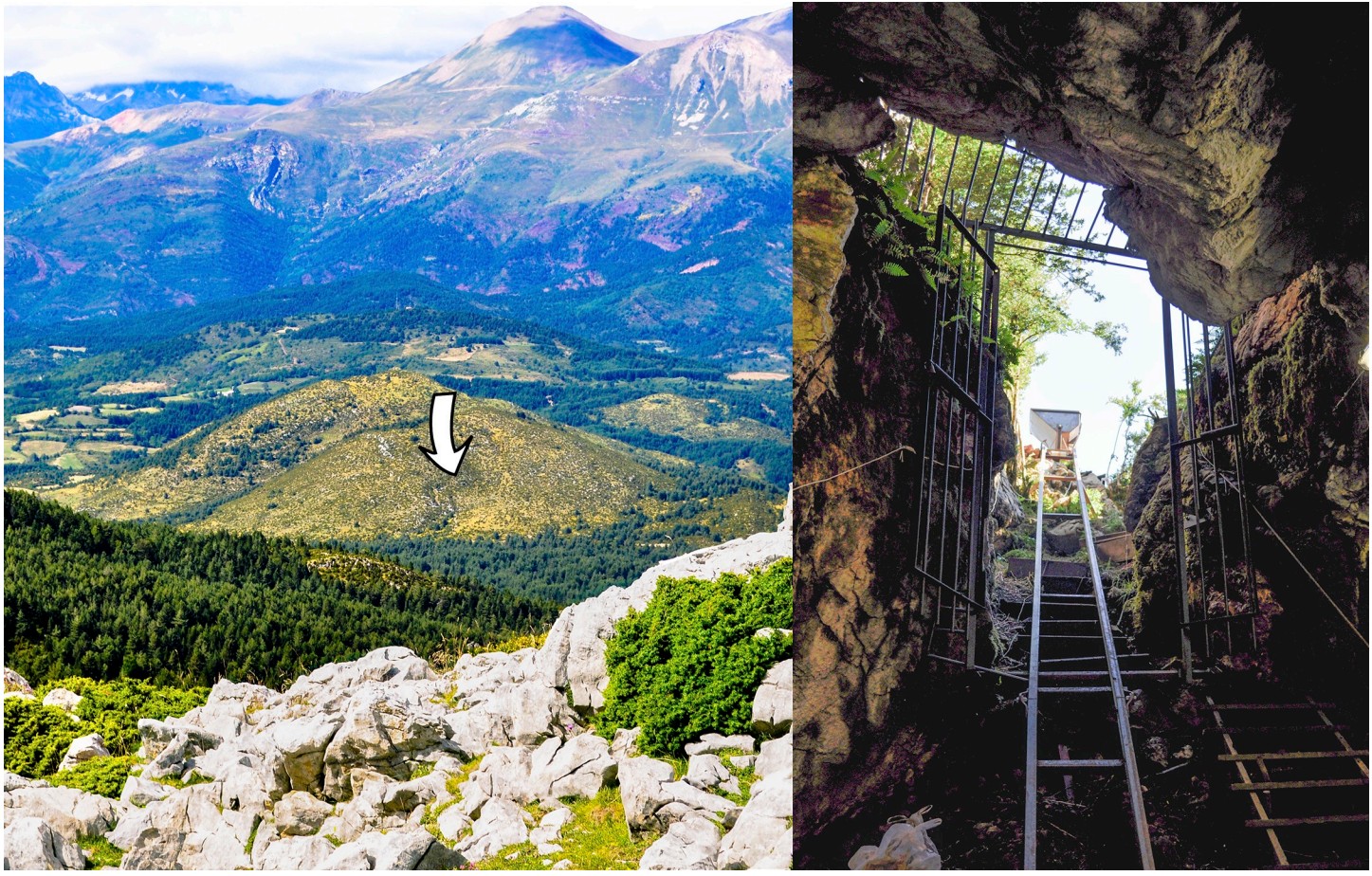

**Fig 1. The archaeological site of Els Trocs.** Left: Location of the cave (pointed by a white arrow) on the southern slope of a karst hill on the high plateau of *Selvaplana;* seen from the pass of the Puerto de las Aras at 1904 m a.s.l. Right: View of the cave entrance from the inside.

Trocs II is represented by a levelling layer, poor in archaeological findings, which was used to regularise the cave floor and on top of which a new pavement made of small and medium-sized stones was arranged (Fig 4).

Associated with this floor three important features were documented [23]: a large hearth standing in the middle of the cave and two dense accumulations of charred and ashed material that are spatially differentiated and derive from burning events which most likely were recurrently performed since Trocs I (Fig 5).

Micromorphological and phytolith analyses concluded that these accumulations do not conform to truly *fumiers* or pen contexts as those recorded in contemporary caves and rock-shelters (*i.e.*, Cova Colomera in the Catalan Pyrenees [34, 35], Cova de les Cendres, Alicante [36], El Mirador in Atapuerca, Burgos [37] or other sites from the Rioja alavesa region in the Basque Country [38]. Dung spherulites were present in modest quantity, demonstrating the occasional presence of livestock inside the cave. Instead, clonocylindrical voids, usually interpreted as excrements of larvae of Adelidae (fairy longhorn moths) and Bibionidae (March flies) feeding on decomposing plant tissues but not on animal detritus [39] were abundant [33]. Thus, the layers of ash and charred material probably originated from periodical cleaning episodes (sweeping and burning) of the grasses used to insulate the occupation floors. Furthermore, the short life cycle of these larvae supports once again the seasonality of the site

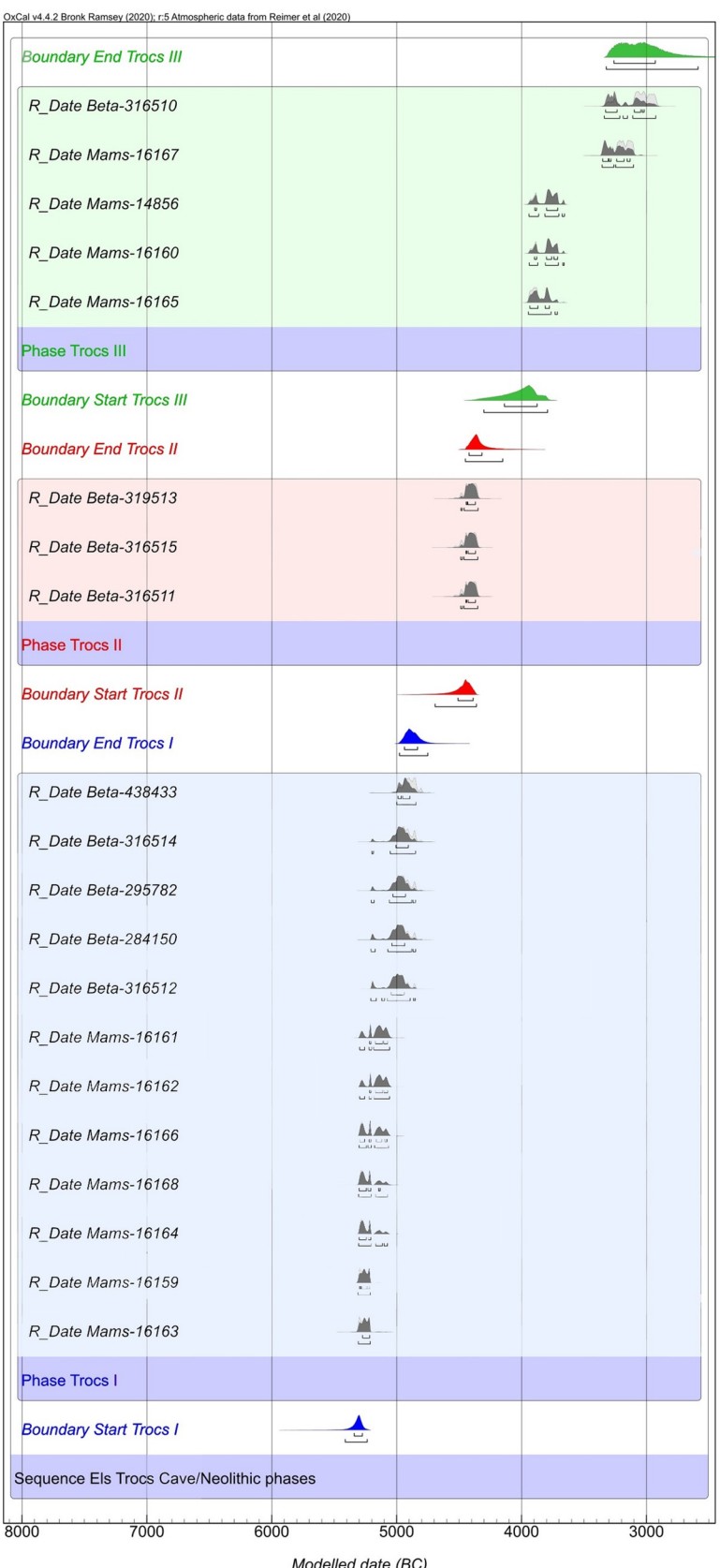

**Fig 2. Bayesian model (phase sequential analysis) for the 14C dates from Els Trocs cave.** Ash grey
curve = calibrated dates; dark grey curve = modelled dates. Boundary estimates for the different Neolithic phases are
presented. The agreement indices of the model and other information about the analysis are included in the S1 Table.

occupation since these insects are most commonly seen in Europe in spring and early summer
[33, 40, 41].

Trocs III comprises the largest sedimentary package (around 0.5 m of thickness). Two fossae yielding some isolated human bones in secondary position and numerous faunal remains
were located at the back of the cave [23]. Also, new hearths were documented. These were successively reutilised until an episode of collapse. The fall of numerous blocks (Fig 5), including
part of the entrance, took place at this moment. Such an event probably explains the abandonment of the use of the cave as a seasonal shelter. Finally, Trocs IV is the uppermost layer characterized by low anthropic activity. Scarce surface findings dated to the Roman period are
mixed with some Neolithic material from the lower layers which were possibly disturbed by
the action of animals in more recent times [23].

## Materials and methods

Permission to carry out the archaeological fieldwork that yielded the datasets used in this
study was granted by the Dirección General de Patrimonio Cultural (Aragón Government,
Spain). The faunal material studied here was mainly recovered from the occupation floors,
pits, fossae and hearths dated to the Early (Trocs I), Middle (Trocs II) and Middle/Late (Trocs
III) Neolithic levels briefly described above. Systematic floatation and wet sieving of the sediments ensured full recovery of macro and small faunal remains. In total, over 19000 vertebrate
specimens (mammal, bird, reptile and amphibian) have been recorded so far (S2 Table). Seventy-nine percent (N = 15120.5) belong to large and medium-sized mammals. Among them,
sixty-three percent (N = 9456.5) were identified to the maximum degree possible (*i.e.*, element,
taxonomic group or animal body-size category), attesting to the good quality preservation at
the site. More than 8500 animal remains deposited in contexts from in-between periods or
located near the entrance of the cave and associated with the upper stratigraphic unit (UE1)
from Trocs III have been excluded from these counts until their dating is defined more
securely.

Bones were refitted when viable, counted, measured when appropriate, and entered into an
IBM SPSS Statistics 25 database. The comparative vertebrate collections of the Archaeobiology
Lab in the Institute of History of the Spanish National Research Council (CSIC) in Madrid
were used for the identifications.

On average, domesticates represent eighty-seven percent of the taxonomically identified
large and medium mammal specimens, clearly outnumbering game and wild species. A total
of 4136 caprine remains were identified. They include 3035 bones, 1068 teeth and 33 horn-
cores. Hence, sheep and goats, followed by suids (morphological and metrical data, currently
under study, suggest that most probably they all belong to wild boar) and cattle may be considered the dominant taxa in the assemblages. In particular, attending to the identified fraction
the caprines account for 90%, 93% and 87% of the husbanded species in Trocs I, Trocs II and
Trocs III, respectively (Table 1).

Due to the lower numbers of suids and wild species of medium size, it may be assumed that
remains recorded under the unidentified medium-sized mammal category primarily belong to
caprines. They include the shafts of long bones, cranial fragments and most vertebrae and ribs.
Therefore, the faunal assemblage from Els Trocs cave is probably one of the richest as regards
the abundance of Neolithic caprine remains in the Iberian Peninsula.

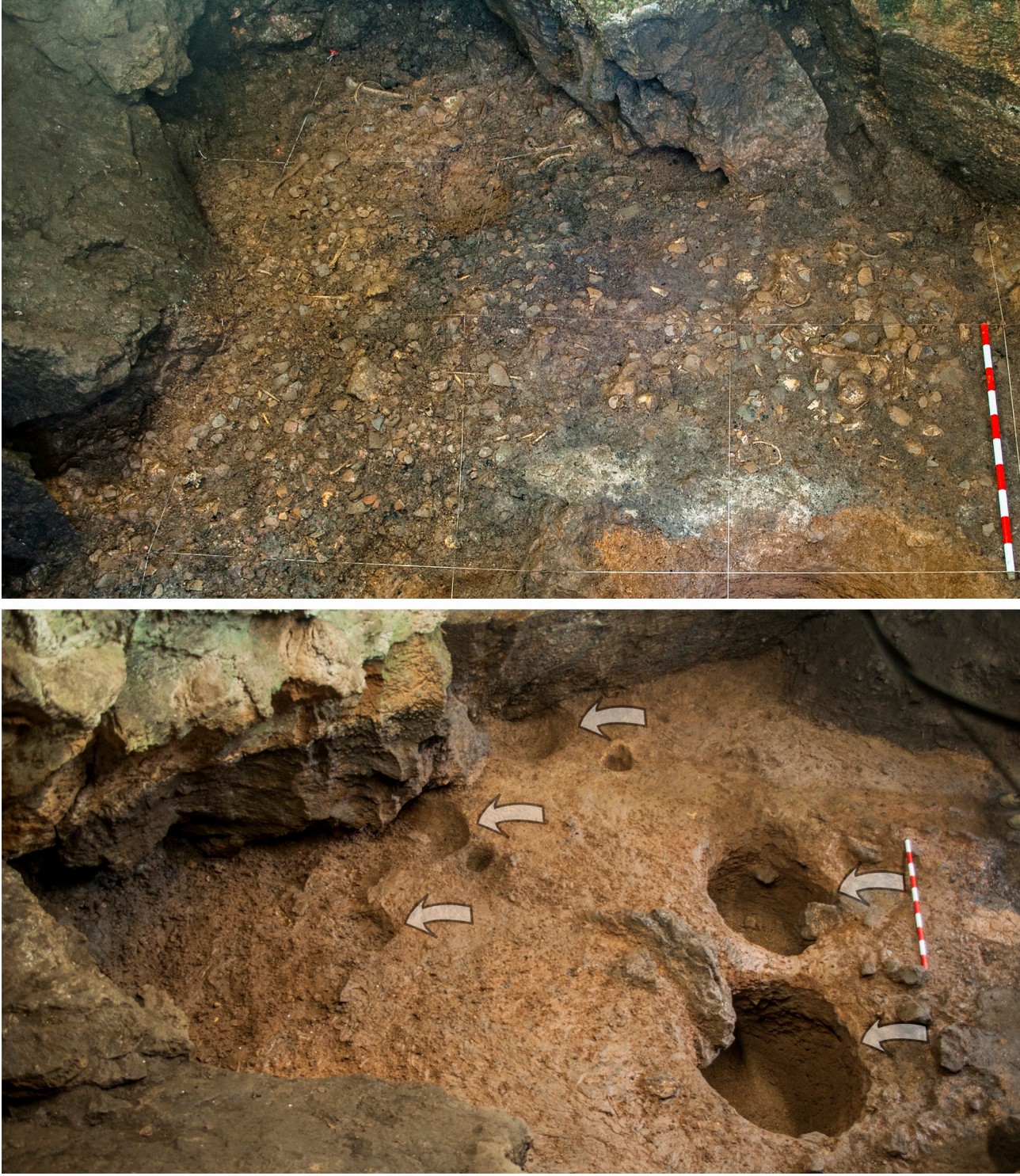

**Fig 3. Archaeological structures from Trocs I.** Top: Floor made of potsherds fragments with scattered faunal and human remains. Bottom: Two of the largest pits excavated in the floor of the cave.

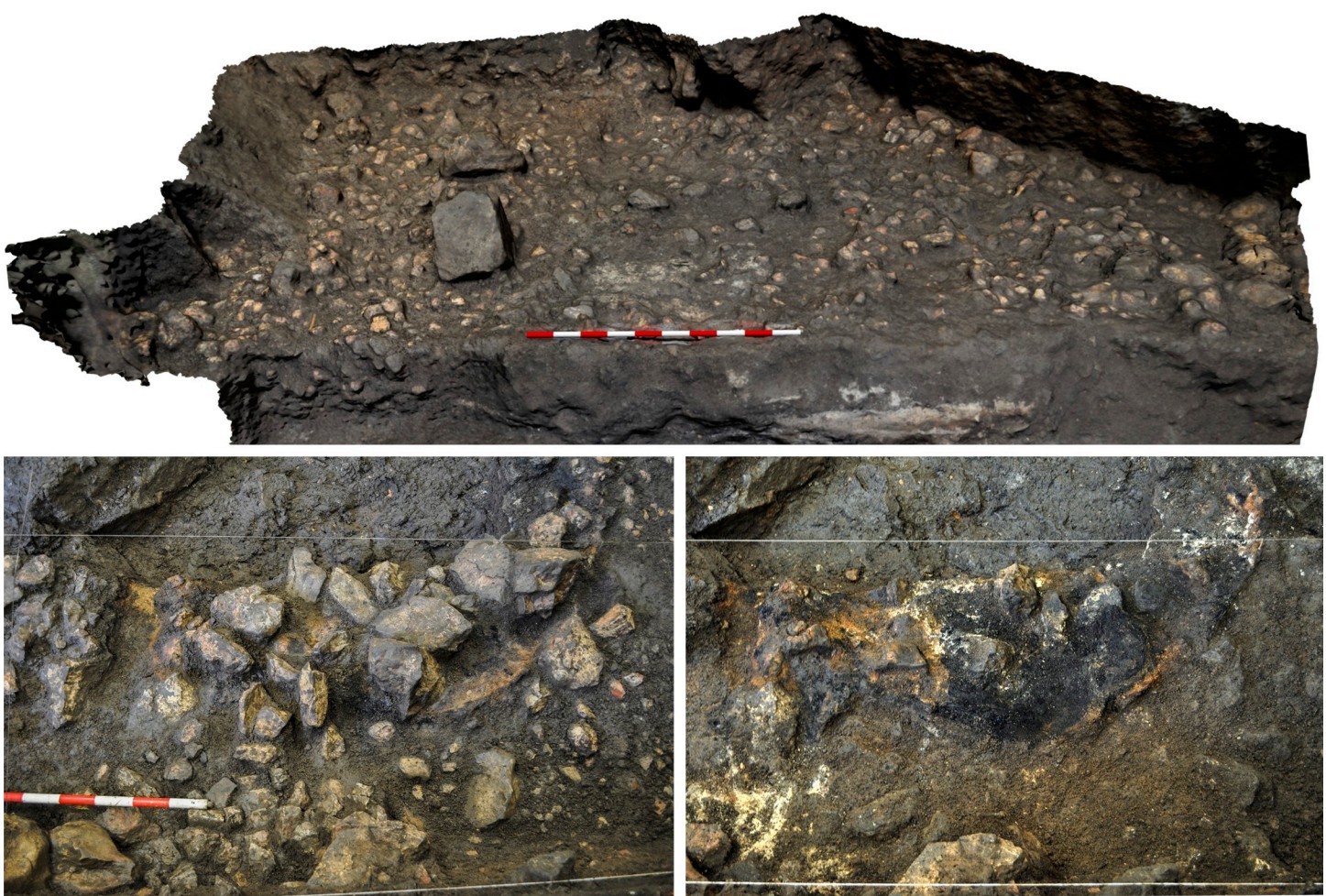

**Fig 4. Archaeological structures from Trocs II.** Top: Detail of the floor fashioned from small and medium-sized stones. Bottom: Different stages of the large and well-preserved hearth documented in the middle of the cave.

Following morphological and metrical criteria [42–47], it was possible to specifically identify part of the sheep and goat bones and teeth. According to the minimum number of elements (MNE), the average sheep:goat ratio in the total assemblage is 14:1, although in Trocs I this is as high as 26:1 (S3 Table). The data for Trocs II (11:1) and Trocs III (10:1) may indicate a slightly more significant contribution of goats to the local flocks in the course of time. Notwithstanding, the relative frequency of sheep clearly proves the heavy reliance on sheep herding of the human groups seasonally visiting the cave. Likewise, the dominance of sheep enables to assume that most of the remains in the mixed sheep/goat category belong to sheep. For the aims of this paper all *Caprinae* specimens are combined and treated as a single analytical unit.

## Estimation of age-at-death patterns

The size and morphological characteristics of many of the osteological immature (with unfused epiphyses) specimens showed the recovery of foetuses, generally in the last weeks of gestation, as well as neonates [48–50]. Given the difficulty in separating these remains, they were binned into the perinatal group. In order to assess the contribution of this age-class in each occupation phase, minimum numbers of cranial (maxilla and mandible) and major front

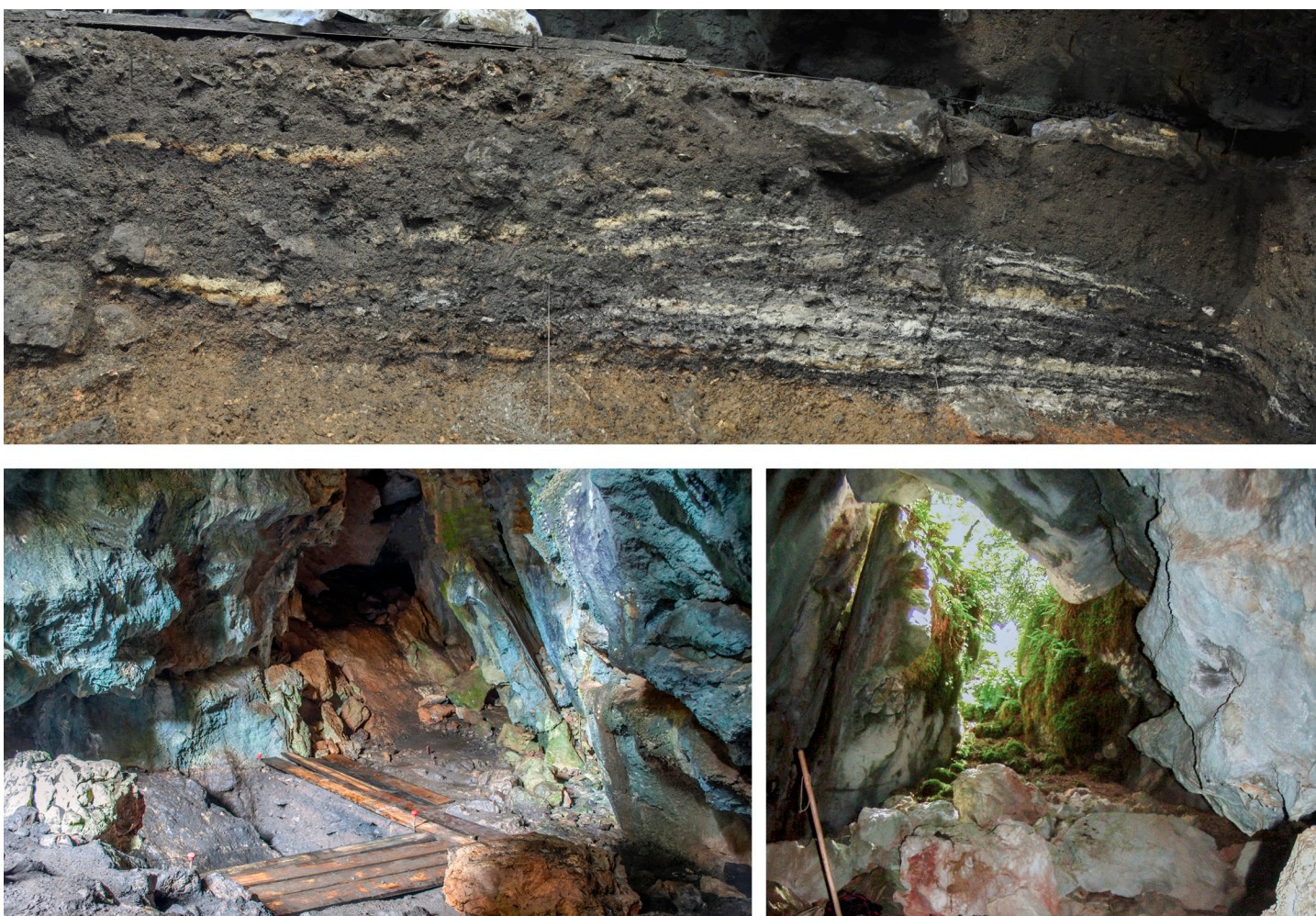

**Fig 5. Archaeological events documented in Trocs II and Trocs III.** Top: The largest ash accumulation with a complex stratigraphic sequence of combustion events. Bottom: View of the numerous large blocks that fell from the roof and walls of the cave, probably forcing the abandonment of the site at the end of Trocs III.

and hind limb bones were counted and their frequency was calculated out of the total caprine MNE in the respective assemblage (S4 Table). MNE were based on the count of diagnostic zones that are unique in each bone. In addition, estimation of mortality patterns was based on the state of limb bone epiphyseal fusion (MNE counts) and mandibular tooth eruption and wear (counts were based on loose deciduous fourth premolars (dP4) or third molars (M3) and mandibles with these teeth plus the first (M1) or the second molar (M2) present).

The fusion sequence and corresponding age ranges followed Zeder [51]. The perinatal group was labelled and its numbers were considered together with the rest of immature specimens to calculate the mortality rate in each age category. Unfused proximal or distal metaphyses were preferably counted to their respective unfused epiphyses. Laterally unfused metapodials in perinatals were counted as half. Rates of survival were estimated from the average number of unfused elements in each of the fusion groups (S1 File). For dental remains, Payne's wear stages and age classes were used [52, 53]. Discrepancies between both methods may show differential preservation and recovery biases affecting dental and bone remains.

**Table 1. Taxonomic spectra for the three Neolithic faunal assemblages recovered in Els Trocs cave using NISP.**

| TAXA | TROCS I | | | TROCS II | | | TROCS III | | | TOTAL | | |
|---|---|---|---|---|---|---|---|---|---|---|---|---|
| | **N** | **%**[b] | **%**[c] | **N** | **%**[b] | **%**[c] | **N** | **%**[b] | **%**[c] | **N** | **%**[b] | **%**[c] |
| Cattle | 179 | 8 | 10 | 94 | 6 | 7 | 174.5 | 11 | 13 | 447.5 | **8** | 10 |
| Sheep | 567 | 27 | | 281 | 18 | | 282.5 | 18 | | 1130.5 | **21** | |
| Goat | 12 | 1 | | 23 | 1 | | 32 | 2 | | 67 | **1** | |
| Sheep/Goat | 1098 | 51 | | 996 | 65 | | 844.5 | 53 | | 2938.5 | **56** | |
| [Sheep+Goat+SG][a] | [1677] | [78] | [90] | [1300] | [85] | [93] | [1159] | [73] | [87] | [4136] | **[79]** | [90] |
| **HUSBANDED TAXA** | **1856** | **87** | **100** | **1394** | **91** | **100** | **1333.5** | **84** | **100** | **4583.5** | **87** | **100** |
| Dog | 1 | <1 | | - | - | | - | | | 1 | <1 | |
| Suids | 192 | 9 | | 121 | 8 | | 223 | 14 | | 536 | **10** | |
| cf. Aurochs | 9 | 2.5 | | 1 | <1 | | - | <1 | | 10 | **1** | |
| cf. Wild goat | 2 | | | 1 | | | 1 | | | 4 | | |
| cf. Chamois | 1 | | | 1 | | | - | | | 2 | | |
| Red deer | 32 | | | 4 | | | 9 | | | 45 | | |
| Roe deer | 10 | | | - | | | 1 | | | 11 | | |
| Hare | 28 | 1 | | 4 | <1 | | 16 | 1 | | 48 | **1** | |
| Rabbit | - | | | 1 | | | 2 | | | 3 | | |
| Bear | - | <1 | | 6 | <1 | | 5 | <1 | | 11 | <**1** | |
| Red fox | 3 | | | 3 | | | 2 | | | 8 | | |
| Wild cat | 1 | | | - | | | - | | | 1 | | |
| **WILD TAXA** | **278** | **13** | | **142** | **9** | | **259** | **16** | | **679** | **13** | |
| **TOTAL IDENTIFIED** | **2135** | **100** | | **1536** | **100** | | **1592.5** | **100** | | **5263.5** | **100** | |
| % IDENTIFIED | 41 | | | 28 | | | 35 | | | 35 | | |
| Large-sized mammal | 296 | | | 144 | | | 244 | | | 684 | | |
| Medium-sized mammal | 1018 | | | 1292 | | | 1199 | | | 3509 | | |
| Undetermined | 1745 | | | 2472 | | | 1447 | | | 5664 | | |
| **TOTAL UNIDENTIFIED** | **3059** | | | **3908** | | | **2890** | | | **9857** | | |
| % UNIDENTIFIED | 59 | | | 72 | | | 65 | | | 65 | | |
| **TOTAL ANALISED** | **5194** | | | **5444** | | | **4482.5** | | | **15120.5** | | |

[a] Total number of *Caprinae* (sheep, goat and sheep/goat) remains.

[b] Relative frequency based on % NISP counts in identified taxa

[c] Relative frequency based on % NISP counts in husbanded taxa only

## Isotopic analyses

Eleven complete or nearly complete hemi-mandibles of sheep were selected for $\delta^{18}O$ and $\delta^{13}C$ stable isotope analyses (S1 File). Isotopic measurements were performed on tooth molars on a seasonal scale using sequential measurements. Oxygen and carbon isotopic compositions are measured conjointly in bioapatite carbonate. Sheep were discriminated from goats according to diagnostic morphological criteria in the third molar (M3) or fourth premolar (P4) [44, 47]. Considering laterality and teeth wear stages they represent different individuals recovered from the occupation sequence. ET Ovis 01, 02, 06 and 12 belong to Trocs I; ET Ovis 03, 04, 05 and 07 derive from Trocs II and ET Ovis 09, 10 and 11 from Trocs III (S5 Table).

All specimens present second molar (M2) crowns already formed. Some of the selected mandibles displayed also the M3 just developed or in an advanced stage of formation. Sheep M2 starts to form at two or three months of age and it is finished around one year of age while the M3 begins to develop at around ten months of age and takes twelve months to form [54, 55]. An estimation of six months is assumed for completion of enamel mineralization [56]. In

both teeth (*i.e.*, unworn crowns), the amount of time taken for the formation of tooth crown covers a period close to one year (M2) or slightly more than one year (M3). This is a period of time desirable in this study, allowing detection of seasonal dynamics throughout a year.

Sampling proceed were performed over the M2 from all specimens. Complementary, the M3 were also sampled in five specimens, totalizing sixteen teeth. After extraction of the teeth, serial or sequential sampling on enamel was performed on the buccal side of the tooth, particularly on the anterior lobe for M2, and the middle lobe for M3. Samples are distanced by intervals of 1 to 1.5 mm covering the whole crown height. Once treated, powdered enamel samples were introduced on a Kiel IV device interfaced to a Delta V Advantage isotope ratio mass spectrometer (IRMS). $\delta^{13}$C and $\delta^{18}$O values are expressed in Vienna-Pee Dee Belemnite (V-PDB) standard (S1 File).

The seasonal reproductive patterns in sheep specimens are also investigated following proposals in Balasse et al. [56, 57] and Tornero et al. [58].

The analytical framework built from our pilot study on sequential sampling molar teeth for isotopic analysis of modern transhumant sheep in Iberia [59] is taken as a baseline to elucidate mobile herding practices of the human groups using Els Trocs cave. The theory and main results of this work, carried out with one of the last flocks that still perform vertical mobility from over winter valley locations of the Ebro river basin up to the central Pyrenees, are summarized in S2 File. It is expected that, if vertical mobility was practiced during the Neolithic occupation of Els Trocs, sampled sheep tooth molars will show a conspicuously similar relationship in their sequential carbon and oxygen series to that observed in the modern reference data set. Summer periods in Els Trocs should be recognized by maximum signals in values of oxygen series and lower values in carbon series. Contrary, the overwinter period in lowland areas should be represented in sequential series by low values in oxygen and higher values in carbon.

## Palaeoparasitological analyses

During the 2014 excavation season fifty sediment samples were collected for palaeoparasitological analyses. Forty-four of them were selected for the present study (13 from Trocs I, 12 from Trocs II and 19 from Trocs III). They come from contexts rich in decomposing organic matter (*i.e.*, containing abundant faunal remains) where the concentration of parasites is expected to be high (S6 Table). Soil samples were treated following the RHM method developed by Dufour and Le Bailly [60] (S1 File). Ten slides of each processed fraction were prepared and examined under an optical microscope (Leica DM-1000 LED) in order to identify parasite biodiversity and concentration along the Neolithic sequence. In addition to palaeoecological information, the analyses will grant valuable data about potential forms of parasitosis as a result of human-animal interactions. From this perspective, they will presumably help to gain some insights into the scale of on-site animal husbandry activities over time.

## Results

### Demography of mortality

Results on perinatal remains abundance show that their relative frequencies depend on the skeletal element counted. Mandibles are always the worst represented together with maxillae and astragali (S4 Table). This suggests that differential preservation and perhaps also recovery biases have negatively affected the fragile and smallest bones of this age group. Thus, if mortality profiles were exclusively based on dental data perinatal mortality would be probably understated. Overall, the average frequency of perinatal remains steadily falls over time: 52% in Trocs I, 44% in Trocs II and 32% in Trocs III (S4 Table).

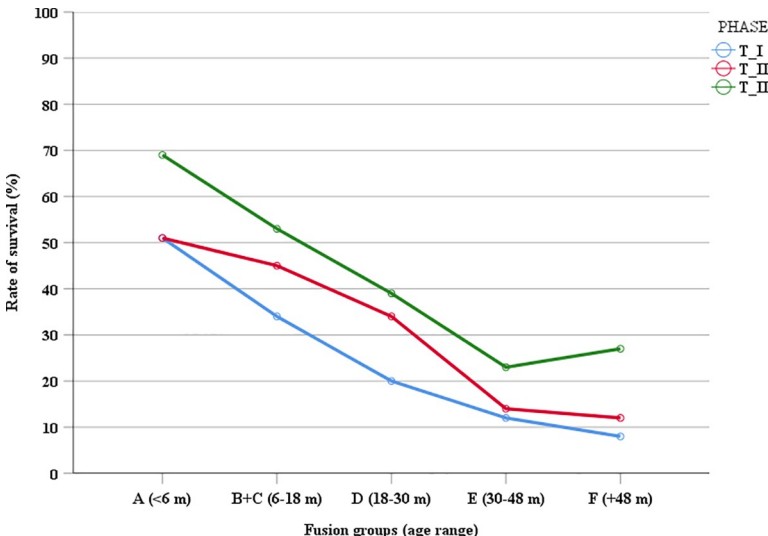

**Fig 6. Survivorship curves for Els Trocs caprines in each occupation period.** The data computed derives from the fusion groups in Table 2.

According to the epiphyseal fusion this situation is mirrored by a progressive increase in the survival rate recorded from 6 to 48 months from Trocs I to Trocs III (Fig 6). During Trocs II and Trocs III fusion stages indicate a slight older age kill-off pattern than in Trocs I, with more animals surviving 18 months of age (B+C: 34%-45%-53%) but also beyond 30 months (D: 20%-34%-39%) and 48 months of age (E: 12%-14%-23%) (Table 2).

The dental wear data, generally regarded as the best proxy for estimating age-at-death [61], shows a similar trend. During Trocs I mortality among the young age classes (A 0–2 months: 20% and B 2–6 months: 17%) represents 37% and the peak of mortality lies in class C (6–12 months: 24%). This means that 61% of the caprines from the Early Neolithic phase died or were culled by their first year of age. With 20% more slaughtered in stage D (1–2 years), only 19% of the animals would have survived into the third year of age. Conversely, in Trocs II and Trocs III, the peak of mortality is delayed to wear stage D (1–2 years, 25% and 31%, respectively) and the last age classes G (4–6 years) and H (6–8 years) exhibit higher frequencies, in particular during Trocs III (15%-4%; Table 3).

In sum, the presence of perinatal remains in all phases evidences that pregnant ewes and does were giving birth during the occupation of the cave, and also that natural or intentional losses occurred. Additionally, the recovery of caprines belonging to Payne's age classes B and C reflects a long-lasting lambing season that spanned over several months. The slaughtering pattern suggests a husbandry strategy orientated to tender meat production in which sheep and goats were preferentially killed before the end of the second year of age. On-site milk exploitation and consumption cannot be excluded but as it will be discussed later on on-site dairying probably remained marginal. Finally, given the seasonal character of the occupations it is plausible that the age-at-death distributions may represent 'truncated' kill-off patterns, reflecting only the part of the management strategy of the farming community associated with the late spring-summer visits to the uplands [62].

## Sheep vertical mobility practices and lambing season

Oxygen ($\delta^{18}O$) and carbon ($\delta^{13}C$) isotopic results obtained from sheep teeth from Els Trocs are presented in S9 Table and summarized in Table 4. Mean $\delta^{18}O$ from all samples was

**Table 2. Culling profile based on epiphyseal fusion data (MNE counts).**

| FUSION GROUP | AGE RANGE | BONE | TROCS I | | | | TROCS II | | | | TROCS III | | | |
|---|---|---|---|---|---|---|---|---|---|---|---|---|---|---|
| | | | P | U | V | F | P | U | V | F | P | U | V | F |
| A | 0–6 m | Px RA | 16 | 1 | - | 18 | 20 | 1 | - | 22 | 6 | 4 | - | 22 |
| | | % unfused | 49 | | | | 49 | | | | 31 | | | |
| B | 6–12 m | Dt HU | 20 | 2 | 2 | 9 | 12 | 1 | 2 | 15 | 9 | 2 | 2 | 15 |
| | | SC | 27 | 12 | - | 16 | 13 | - | 2 | 10 | 16 | 1 | - | 19 |
| | | % unfused | 69 | | | | 47 | | | | 44 | | | |
| C | 12–18 m | Px PH2 | 9 | 2 | 2 | 4 | 2 | 3 | 2 | 1 | 1 | - | 2 | 5 |
| | | Px PH1 | 14 | 5 | 4 | 9 | 4 | 7 | 1 | 1 | 4 | 8 | - | 4 |
| | | % unfused | 61 | | | | 76 | | | | 54 | | | |
| B+C | | | 66 | | | | 55 | | | | 47 | | | |
| D | 18–30 m | Dt TI | 23 | 5 | - | 13 | 14 | 5 | - | 14 | 10 | 5 | - | 18 |
| | | Dt MTP | 38 | 12 | - | 7 | 14 | 5 | - | 6 | 15 | 10 | - | 8 |
| | | % unfused | 80 | | | | 66 | | | | 61 | | | |
| E | 30–48 m | CAL | 10 | 3 | - | 3 | 5 | 2 | - | 4 | 3 | 4 | - | 3 |
| | | Px FE | 17 | 2 | 1 | 2 | 9 | 1 | - | 1 | 6 | 2 | - | 2 |
| | | Dt FE | 17 | 5 | 1 | - | 11 | 3 | - | 1 | 5 | 3 | 1 | 4 |
| | | Px UL | 11 | 1 | 2 | 1 | 10 | 6 | - | 1 | 7 | 4 | - | 1 |
| | | Dt RA | 16 | 9 | - | 3 | 21 | 7 | 1 | 2 | 5 | 8 | - | 3 |
| | | Px TI | 22 | 2 | 1 | 2 | 8 | 5 | 2 | 2 | 6 | 8 | 2 | 2 |
| | | % unfused | 88 | | | | 86 | | | | 77 | | | |
| F | + 48 m | Px HU | 20 | 4 | 1 | 1 | 12 | 3 | - | 2 | 9 | 2 | 3 | 1 |
| | | % unfused | 92 | | | | 88 | | | | 73 | | | |
| | | TOTAL | 260 | 65 | 14 | 88 | 155 | 49 | 10 | 82 | 102 | 61 | 10 | 107 |
| % unfused | | | 76 | | | | 69 | | | | 58 | | | |
| Total MNE | | | 427 | | | | 296 | | | | 280 | | | |

Abbreviations: P: perinatal; U: unfused; V: fusing; F: fused; Px: proximal; Dt: distal; RA: radius; HU: humerus; SC: scapula; PH2: second phalanx; PH1: first phalanx; TI: tibia; MTP: metapodials; CAL: calcaneus; FE: femur; UL: ulna.

-2.1 ± 0.83 ‰, ranging from 2.3 ‰ to -4.9 ‰. Mean $\delta^{13}C$ value from all samples was -11.4 ± 0.36 ‰, ranging from -9.6 ‰ to -12.3 ‰. In M2s, mean $\delta^{18}O$ value was -1.9 ±1.25 ‰ (minimum value -4.7 ‰; maximum value 2.3 ‰) and mean $\delta^{13}C$ value was -11.5 ±0.51 ‰ (minimum value -12.3 ‰; maximum value -9.6 ‰). In M3s, mean $\delta^{18}O$ value was -2.5 ±1.03 ‰ (minimum value -4.9 ‰; maximum value -0.4 ‰) and mean $\delta^{13}C$ value was -11.3 ±0.31 ‰ (minimum value -12.3‰; maximum value -10.8‰).

Sequences of $\delta^{18}O$ and $\delta^{13}C$ values for each individual M2 are shown in Fig 7.

In all specimens, oxygen and carbon sequences show high intra-tooth variation. This variation follows a sinusoidal pattern and amplitude of variation is higher in $\delta^{18}O$ than in $\delta^{13}C$ sequences. In both sequences (and in all teeth) these distributions present clear maximum and minimum peaks events, reflecting seasonal fluctuations occurring during the formation of the crown sequence. The $\delta^{13}C$ values measured in all sheep teeth indicate a diet widely based on $C_3$ plants throughout the year. Further, our recent study on modern transhumant sheep in the area (S2 File) enables to suggest that the variation in $\delta^{13}C$ values along the tooth reflecting changes in the sheep diet may derive from the differences in prevailing growing conditions of ingested $C_3$ plants in the lowlands and highlands and, in less proportion, from the contribution of $C_4$ plants. Likewise, the sinusoidal variation observed in the serial series of $\delta^{18}O$ values (mainly derived from ingested water) relates to the seasonal climatic cycle, with the highest

**Table 3. Kill-off patterns based on mandibular eruption and wear stages.**

|  | WEAR STAGES | AGE | TROCS I N | TROCS I % | TROCS II N | TROCS II % | TROCS III N | TROCS III % |
|---|---|---|---|---|---|---|---|---|
| IMMATURE | A | 0–2 m | 9 | 20 | 7 | 13 | 2 | 4 |
|  | B | 2–6 m | 8 | 17 | 10 | 19 | 1 | 2 |
|  | C | 6–12 m | 11 | **24** | 11 | 21 | 12 | 25 |
|  | *Subtotal* |  | *28* | *61* | *28* | *54* | *15* | *31* |
| SUB ADULT | D | 1–2 y | 9 | 20 | 13 | **25** | 15 | **31** |
|  | E | 2–3 y | 3 | 6 | 2 | 4 | 4 | 8 |
|  | *Subtotal* |  | *12* | *26* | *15* | *29* | *19* | *40* |
| ADULT | F | 3–4 y | 2 | 4 | 2 | 4 | 5 | 10 |
|  | G | 4–6 y | 3 | 6 | 7 | 13 | 7 | 15 |
|  | H | 6–8 y | 1 | 2 | - | - | 2 | 4 |
|  | I | +8 y | - | - | - | - | - | - |
|  | *Subtotal* |  | *6* | *13* | *9* | *17* | *14* | *29* |
| TOTAL |  |  | 46 | 100 | 52 | 100 | 48 | 100 |

The percentages in bold indicate the age class representing the peak of mortality.

values occurring during the warmest months and the lowest during the coldest months. Thus, the low $\delta^{13}C$ values and high $\delta^{18}O$ values along the crown sequences may be signaling at grazing in the $C_3$ plant dominated high mountain areas during the summer. Conversely, during the overwinter period, when $\delta^{18}O$ signatures show its lowest values, sheep grazing in the valley locations would have ingested $\delta^{13}C$-enriched $C_3$ plants or/and mixed $C_3/C_4$ plants, resulting in higher $\delta^{13}C$ values than during the summer period. In this way, the described co-variation of oxygen and carbon isotope measurements helps to clarify seasonal vertical mobility in the sequentially sampled teeth.

**Table 4. Isotopic results ($\delta^{18}O$ and $\delta^{13}O$ values) in bioapatite samples from Els Trocs cave sheep lower second (M2) and third (M3) molars.**

|  | Tooth | n | $\delta^{18}O_{V-PDB}$ (‰) Max | Min | Mean | Range | $\delta^{13}C_{V-PDB}$ (‰) Max | Min | Mean | Range |
|---|---|---|---|---|---|---|---|---|---|---|
| ET Ovis 01 | M2 | 19 | -0.6 | -4.7 | -2.8 | -4.1 | -11.3 | -12.0 | -11.7 | -0.7 |
|  | M3 | 10 | -0.4 | -2.6 | -1.3 | -2.2 | -11.0 | -11.3 | -11.2 | -0.4 |
| ET Ovis 02 | M2 | 15 | 2.3 | -2.5 | -0.7 | -4.8 | -9.6 | -11.3 | -10.4 | -1.8 |
| ET Ovis 03 | M2 | 17 | -0.6 | -3.8 | -1.9 | -3.2 | -11.1 | -12.3 | -11.7 | -1.2 |
|  | M3 | 10 | -1.8 | -3.2 | -2.6 | -1.4 | -11.3 | -12.0 | -11.6 | -0.7 |
| ET Ovis 04 | M2 | 15 | 0.9 | -1.8 | -0.4 | -2.6 | -10.6 | -12.3 | -11.3 | -1.7 |
|  | M3 | 17 | -1.4 | -2.8 | -2.4 | -1.4 | -10.8 | -12.3 | -11.3 | -1.5 |
| ET Ovis 05 | M2 | 17 | -1.8 | -3.6 | -2.5 | -1.8 | -11.3 | -12.2 | -11.6 | -0.9 |
| ET Ovis 06 | M2 | 18 | -1.2 | -3-6 | -2.2 | -2.4 | -11.5 | -12-1 | -11.8 | -0.6 |
|  | M3 | 16 | -1.4 | -3.1 | -2.2 | -1.7 | -11.2 | -11.9 | -11.5 | -0.7 |
| ET Ovis 07 | M2 | 14 | 0.6 | -3.0 | -1.4 | -3.6 | -11.3 | -11.7 | -11.5 | -0.4 |
| ET Ovis 09 | M2 | 14 | -0.2 | -2.9 | -1.9 | -2.7 | -11.7 | -12.3 | -12.0 | -0.6 |
| ET Ovis 10 | M2 | 16 | -1.2 | -3.3 | -2.2 | -2.1 | -11.2 | -11.6 | -11.4 | -0.4 |
| ET Ovis 11 | M2 | 13 | -1-9 | -4.0 | -2.6 | -2.1 | -10.7 | -11.5 | -11.1 | -0.8 |
|  | M3 | 19 | -2.0 | -4.9 | -3.6 | -2.9 | -10.8 | -11.6 | -11.3 | -0.8 |
| ET Ovis 12 | M2 | 9 | -2.4 | -3.2 | -2.9 | -0.8 | -11.2 | -11.8 | -11.5 | -0.6 |

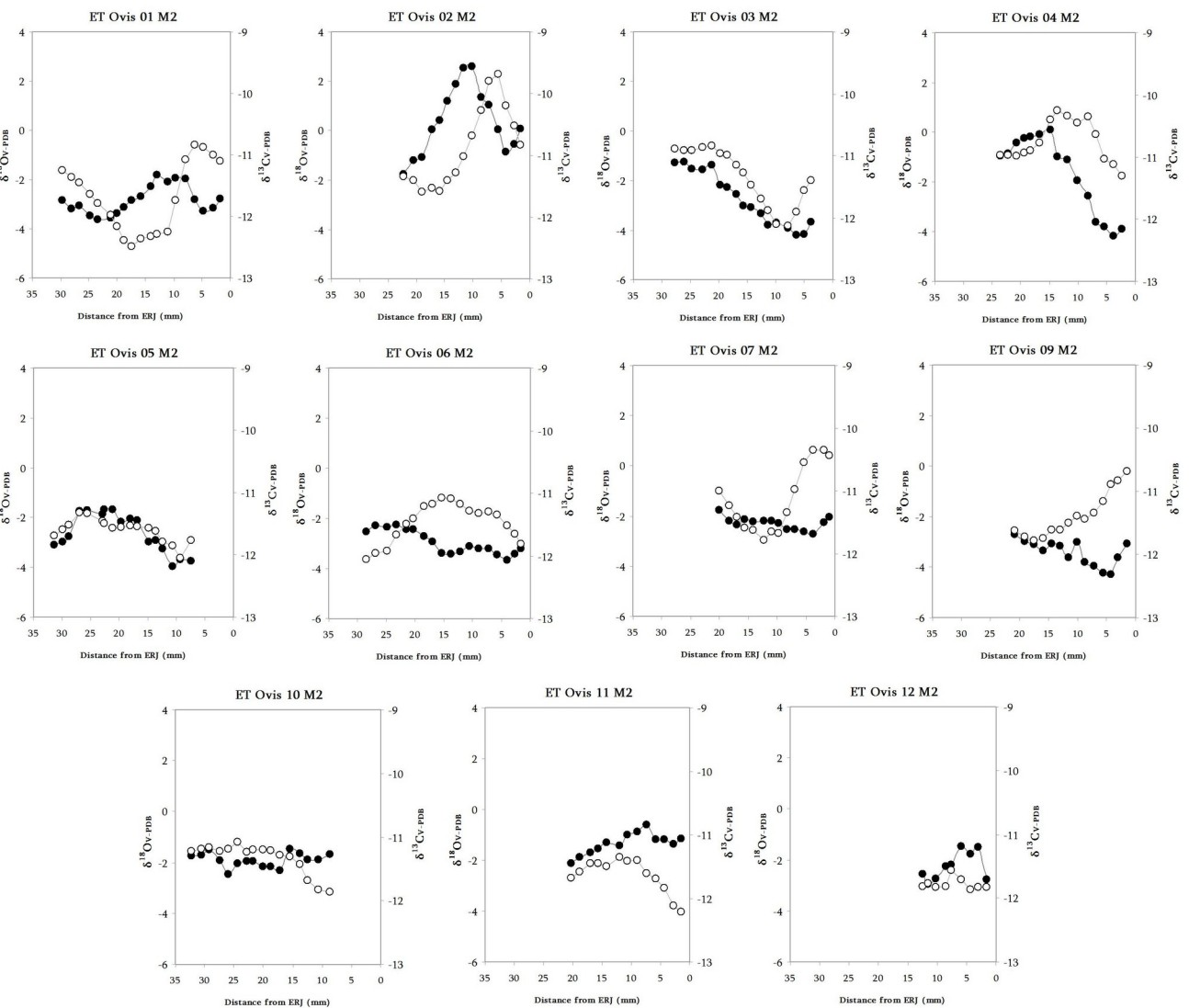

**Fig 7. Results from the sequential analysis of oxygen (δ¹⁸OV-PDB, white circles) and carbon (δ¹³CV-PDB, black circles) isotope composition in sampled enamel bioapatite from Cova de Els Trocs sheep lower second molars (M2).**

The way the encompassed oscillation of the $\delta^{18}O$ and $\delta^{13}C$ sequences occurs in the sampled M2 is variable between the specimens. In ET Ovis 07, 09 and 10 $\delta^{18}O$ and $\delta^{13}C$ sequences looks like inverse and quite similar, although not identical, to that observed in the encompassed $\delta^{18}O$ and $\delta^{13}C$ sequences retrieved from modern transhumant sheep in the area [59], indicating that these individuals were involved in vertical mobility during formation of M2. ET Ovis 01, 02, 04, 06 and 11 just show a non-encompassed pattern while ET Ovis 03 and 05 show a clearly parallel relationship on its isotopic sequences, suggesting a sedentary life history during the formation of the M2. Finally, ET Ovis 12 could not be evaluated due to its short crown.

In order to better understanding the cases in which the inverse relationship between carbon and oxygen was not clear, available M3s (ET Ovis 01, 04, 06 and 11) were analysed, too. Additionally, it was decided to examine the M3 of specimens with non-vertical mobility histories in M2 to check if such pattern changed later during their life. ET Ovis 03 was the only specimen

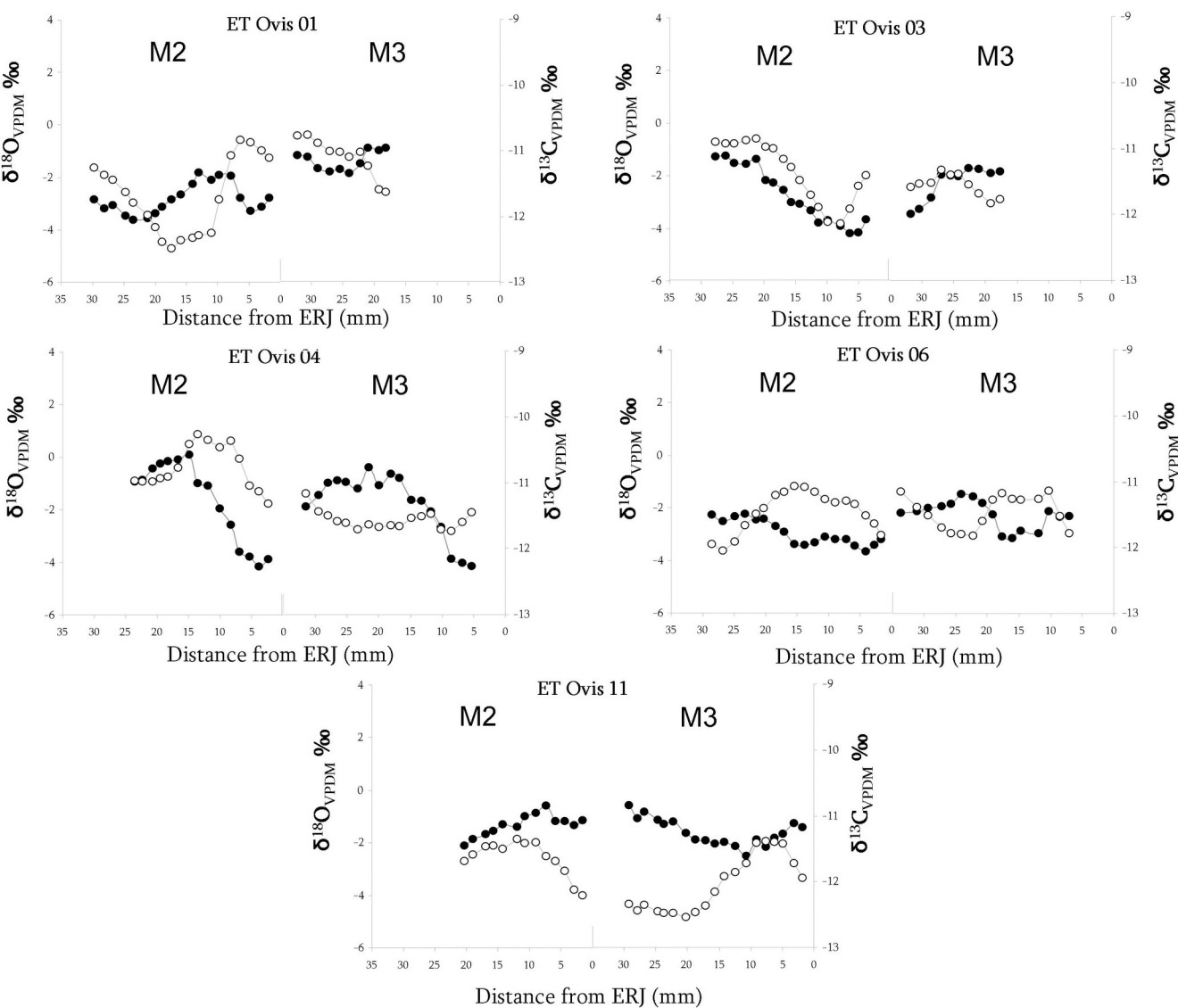

**Fig 8. Sequential analysis of oxygen (δ¹⁸O_V-PDB, white circles) and carbon (δ¹³C_V-PDB, black circles) isotope composition in lower second (M2) and third (M3) molars in five of the sheep samples from Els Trocs cave.**

from this last group that could be analysed because the M3 from ET Ovis 05 still was not totally formed when the animal died. In all sampled M3, encompassed oscillation of the δ¹⁸O and δ¹³C sequences show a non-similar relationship or, directly, inverse relationship, suggesting the existence of vertical movements during the formation of this tooth (Fig 8).

Furthermore, the combination of both isotopic sequences helped to solve the uncertain readings of isolated M2 sequences. Vertical mobility was not practiced by specimens ET Ovis 01, 04, 06 and 11 at the beginning of the formation of M2 (*i.e.*, carbon and oxygen sequences show a parallel relationship at the beginning of the isotopic sequence), but this scenario shifts to an inverse pattern later on, when formation of M2 is advanced and throughout all the sequence obtained in M3.

Hence, these accurate temporal analyses indicate that these animals were not involved in vertical movements during their early age stages (represented by the early steps of formation

of M2 crowns), but they did it later on during their lives. Indeed, this observation seems to be present also in those specimens, although in a lesser degree, where vertical movements were recognized while considering only M2 sequences (ET Ovis 07 and 09). In all of them, first carbon and oxygen values (*i.e.*, 2–3 first samples) present a parallel relationship which quickly shifts to an inverse pattern. Long away from these last observations is ET Ovis 03. In this sheep, M2 sequences show a parallel relationship but an inverse relationship was recognized in the advanced M3 sequence, suggesting that this animal, contrary to the previous cases, did not practice vertical movements until a much older age.

In brief, after evaluating all available results in M2 and M3, it has been possible not only to positively identify vertical mobility in most of the analysed sheep specimens from Els Trocs but also to confirm that these movements occurred along the three occupation phases. Specimens ET Ovis 01 and 06 from Trocs I show clear evidence of having practiced altitudinal movements during the advanced formation of the M2 and throughout all the M3. In Trocs II, three out of the four sample specimens were involved in this pastoral strategy although at different stages in their lives. ET Ovis 07 went to the uplands during formation of M2, ET Ovis 04 during the advanced formation of the M2 and throughout all the M3, while ET Ovis 03 was engaged in vertical mobility only during the formation of the M3. Finally, during Trocs III, vertical mobility is well documented in ET Ovis 09 and 10, both during the formation of the M2. Moreover, the recognition of an inverse pattern in the $\delta^{18}$O and $\delta^{13}$C sequences of M2 and M3 belonging to the same individual points to a systematic practice of this pastoral activity over different consecutive years.

Another interesting issue relates to the different ages at which the sampled sheep begin to practice vertical movements. With the aim to gain a better understanding on this matter, seasonal reproductive patterns (*i.e.*, lambing season) were also investigated in the same specimens following proposals in Balasse et al. [56, 57] and Tornero et al. [58].

We use the $\delta^{18}$O sequences observed in all sampled M2, except ET Ovis 12 due to its short sequence (S5 Table). Sinusoidal variation in the $\delta^{18}$O values recorded in a tooth reflects the seasonal climatic cycle. Position of the oxygen optimum values (maximum or minimum seasonal events) gives information about the period of enamel formation (*i.e.*, the season and seasonality of birth) [58, 63, 64].

The Els Trocs dataset shows that the highest $\delta^{18}$O values in the M2 sequences occur at different distance (in mm) from the enamel/root junction (ERJ): from 0 to 5 mm in ET Ovis 07 and 09; from 5 to 10 mm in ET Ovis 01 and 02; from 10 to 15 mm in ET Ovis 04, 06 and 11, and from 20 to 30 mm in ET Ovis 03, 05 and 10 (Fig 9). Such inter-individual variability suggests important differences in the season of birth within the specimens analyzed.

In order to quantify the inter-individual variability in the positioning in tooth crown of the maximum $\delta^{18}$O value, the oxygen isotope sequences were modelled using an equation derived from a cosine function [56]. Unfortunately, the modelling procedure cannot be applied in those sequences where minimum and maximum $\delta^{18}$O values are not well-identified. In our case, this excludes ET Ovis 04, 06, 09 and 11. Normalized data sets (i.e. $x_0/X$) are then compared with those obtained in modern sheep populations with known date of birth, allowing the estimation of the differences in lambing season within the specimens analyzed (S7 Table and Fig 10).

Results show that in Trocs I lambing season occurred in mid-late spring. During Trocs II, early spring birth is attested in one sheep (ET Ovis 07) but two other specimens fall in the mid-autumn group. The only specimen available from Trocs III was born also in autumn (Fig 10). In sum, our data suggest a shift toward a more variable lambing season from the Early (Trocs I) to the Middle Neolithic (Trocs II), highlighting the occurrence of autumn births and the continuation of such breeding practice over time (Trocs III).

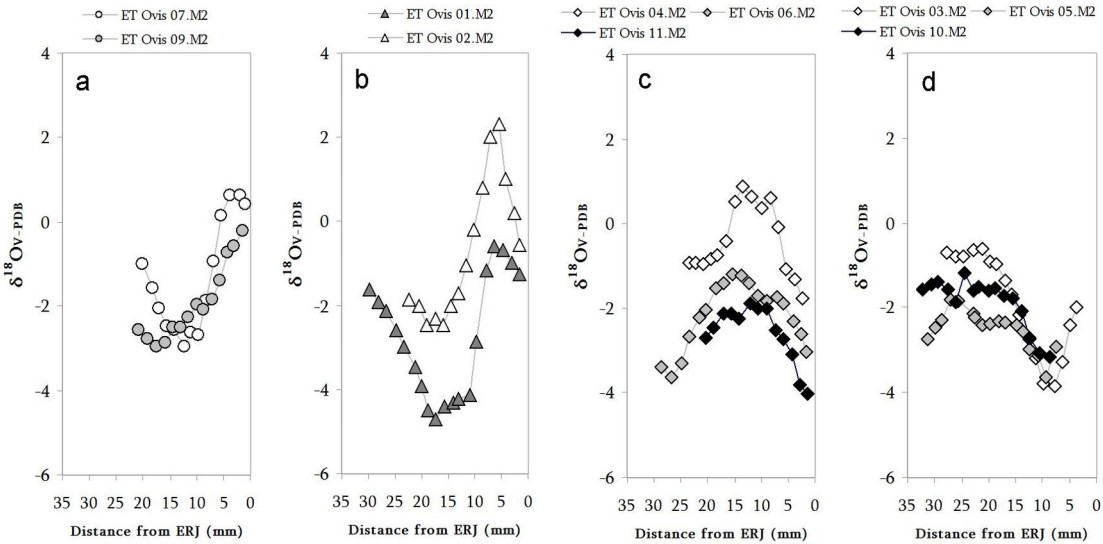

**Fig 9. Sequential δ¹⁸O$_{V-PDB}$ values in all second molars, classified according to the position in tooth crown of the maximum oxygen event in the tooth crown.** Type a: max. δ¹⁸O$_{V-PDB}$ values between 0 to 5mm; Type b: max. δ¹⁸O$_{V-PDB}$ values between >5 to 10 mm; Type c: max. δ¹⁸O$_{V-PDB}$ values between 10 to 15mm; and Type d: max. δ¹⁸O$_{V-PDB}$ values between 20 to 30mm.

## Diachronic spectrum and concentration of parasites

Eggs of various helminth worms, including roundworms (*Nematodes*) and flatworms (*Trematodes*), which may parasitize the human or animal digestive tract, were found in twenty-four out of the forty-four sediment samples analysed. Given the constant humidity (*ca.* 90%) and temperature (8˚-9˚C) inside the cave, the eggs recovered were very well preserved. The taxonomic composition (Fig 11) and frequency distribution of their number suggest a clear increase in the abundance and diversity of parasites throughout the Neolithic sequence. Whereas positive samples in Trocs I are rather weak (31%) and little diverse (there are just three different taxa: *Capillaria* sp., *Trichuris* sp. and *Dicrocoelium* sp.), in Trocs II they attain

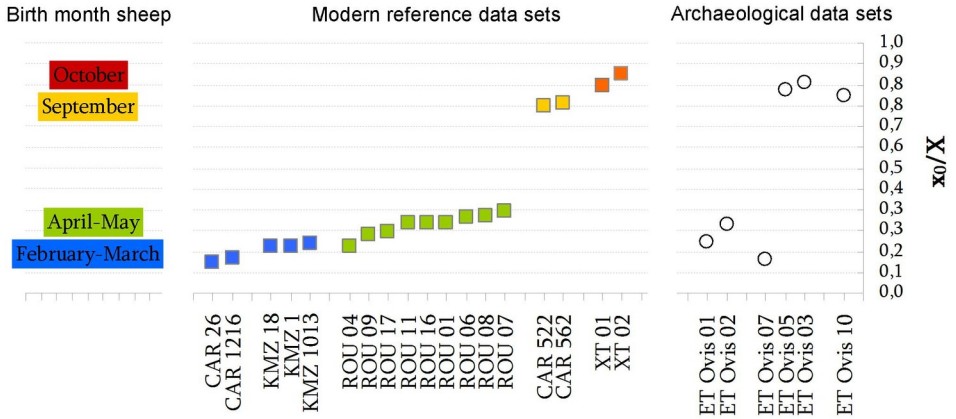

**Fig 10. Normalized data sets (*x₀/X*) from modelled δ¹⁸O M2 of sheep from Els Trocs cave and comparison with data from modern reference populations with known date of birth.** Specimens CF Ovis 0026 & 1216 were born between late January to early February whereas CF Ovis 522 & 562 were born in middle September [65]; ROU 01, 04, 06, 07, 08, 09, 11, 16 & 17 were born between late April and early May [56]; and XT Ovis 01 & 02 were born in early October [59].

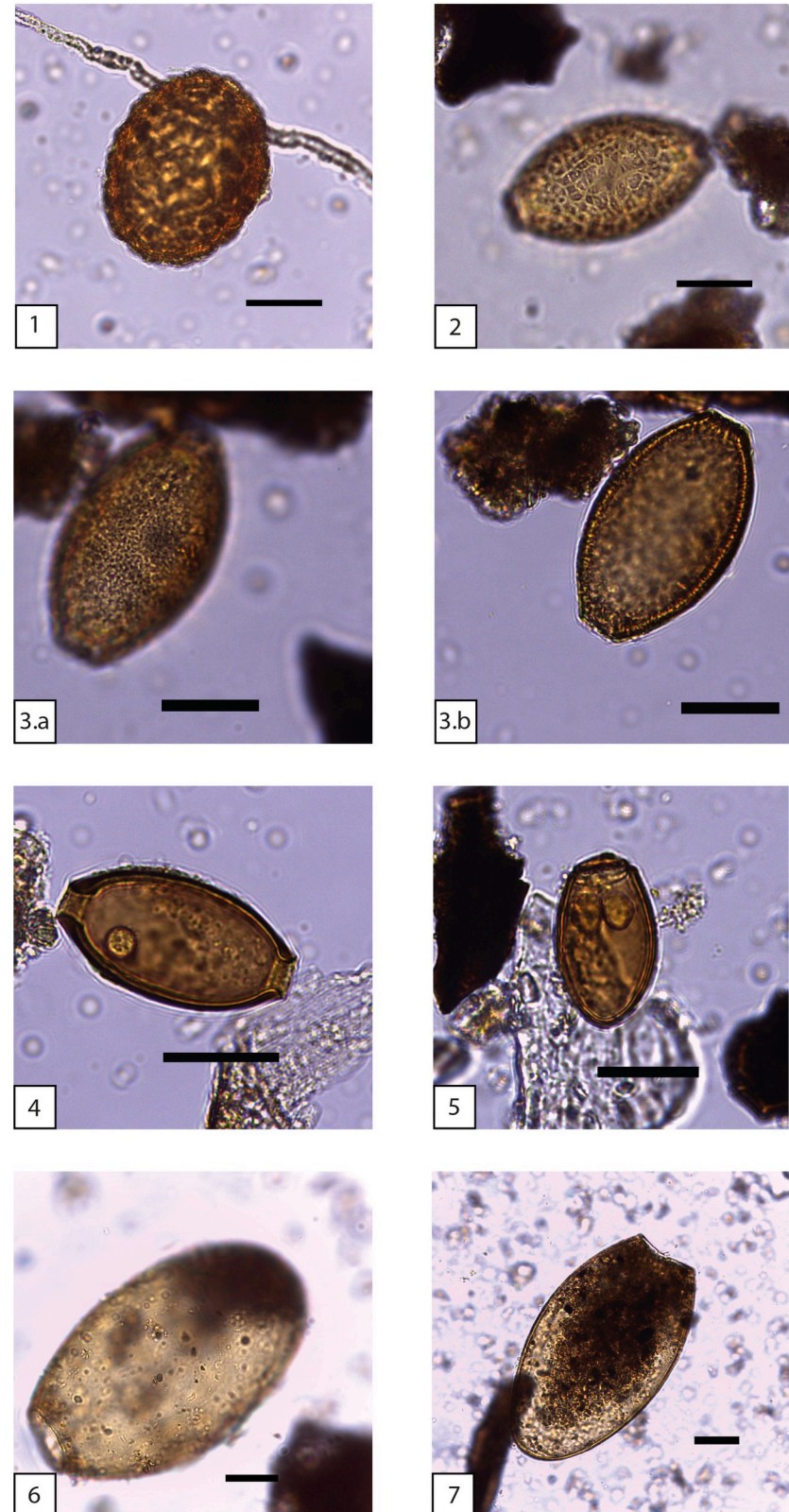

**Fig 11. Helminth eggs identified in the sediments from Els Trocs cave.** *Ascaris* sp. (n. 1), *Capillaria* sp. (n. 2, 3.a & 3. b), *Trichuris* sp. (n. 4), *Dicrocoelium* sp. (n. 5), *Paramphistomum* sp. (n. 6) and *Fasciola* sp. (n. 7). Scale: 25 micra.

**Table 5. Results of positive microscopy examinations in the parasitological samples of Els Trocs cave by chronological phase.**

|  | All samples (*n* = 44) | % | 95% CI * | Trocs I (*n* = 13) | Trocs II (*n* = 12) | Trocs III (*n* = 19) |
|---|---|---|---|---|---|---|
| **Overall positivity rate** ** | **24** | **54.55** | **40.07–68.29** | **4** | **6** | **14** |
| **Nematodes** |  |  |  |  |  |  |
| *Ascaris* sp. | 9 | 20.45 | 11.15–34.50 | 0 | 3 | 6 |
| *Capillaria* sp. | 13 | 29.55 | 18.16–44.22 | 1 | 2 | 10 |
| *Trichuris* sp. | 11 | 25 | 14.57–39.44 | 2 | 1 | 8 |
| **Trematodes** |  |  |  |  |  |  |
| *Dicrocoelium* sp. | 21 | 47.73 | 33.75–62.06 | 3 | 6 | 12 |
| *Paramphistomum* sp. | 5 | 11.36 | 4.95–23.98 | 0 | 1 | 4 |
| *Fasciola* sp. | 3 | 6.82 | 2.35–18.23 | 0 | 0 | 3 |

* 95% confidence interval

** Presence of eggs from one helminth genus or more per soil sample.

50%. Also, each positive sample has at least one parasite egg. Concentrations of the previously identified genera are slightly higher and new genera such as *Ascaris* and *Paramphistomum*, are also found. Finally, Trocs III is the richest phase in positive samples (74%), biodiversity of parasites and number of recovered remains. Besides the helminthes already noted, a new taxon exclusively linked with animals, *Fasciola*, is recorded (Table 5).

A Fisher's exact test confirms the statistically significance ($p < 0.05$) of the increase in helminthiases between Trocs I and Trocs III as well as a higher density of *Capillaria* sp. and *Dicrocoelium* sp. eggs in Trocs III (S8 Table). However, due to the relatively low number of positive results (overall rate under 55%) and because the epidemiology of most helminthiases former to the last two or three centuries is unknown, care should be taken when inferring biological data from the identified parasites and associated definitive or intermediate hosts. While *Ascaris* and *Trichuris* have a direct life cycle involving fecal–oral transmission of infective eggs, certain worms of the genera *Capillaria*, *Dicrocoelium* and *Fasciola* are recognized as agents of pure zoonotic diseases.

In Europe, roundworms of the genus *Ascaris* comprise two species whose divergence has recently been claimed to occur in the early Neolithic [66]. *Ascaris suum* (Goeze, 1782) is the helminth parasite of suids and *Ascaris lumbricoïdes* (Linné, 1758) has humans as natural reservoir. Because their eggs are virtually indistinguishable when viewed under light microscopy, the remains from Trocs II and Trocs III potentially derive from faeces of either host and/or the offal remains of infected suids that were processed inside the cave. The genus *Trichuris* (Roederer, 1761) comprises more than twenty species that may infect more than 60 mammalian taxa [67]. *Trichuris trichiura* is the only one specific for human hosts. This parasite most likely originated in Africa and after its introduction to Asia, maybe by *Homo erectus*, it shifted hosts to pigs [68]. The other Nematode genus identified is *Capillaria*. Among the thirteen *Capillaria* eggs it was possible to differentiate two types according to their morphology and biometry (Fig 11). Type 1 comprises medium-sized eggs (64,8μm x 36,3μm) and Type 2 includes slightly smaller eggs (55,5μm x 31,7μm). The striated morphology of the shells makes them compatible with *Capillaria hepatica* (Brancroft, 1893). However, as the subfamily Capillariinae includes about 300 species of nematodes parasitizing all groups of vertebrates and the ecology, biology and specific definitive hosts are presently known only for 25 species [69] no specific identification was attempted.

*Dicrocoelium dendriticum* or "lancet liver fluke" and *Fasciola hepatica*, also known as "common liver fluke" or "sheep liver fluke", are the only species of these Trematode genera

currently present in Europe [70, 71]. *Dicrocoelium* mainly infects ruminants, in particular sheep [72]. Human infection is very rare because there are two intermediate hosts in its life-cycle, a terrestrial snail and the ant *Formica fusca* [73], but this does not mean it could not have happened in Neolithic times. *Fasciola* may parasitize the liver of any mammal including humans but sheep and cattle are preferred hosts. Interestingly, the ecological criteria for the natural development of both genera are different. Whilst *Dicrocoelium* sp. lives in dry pastures [74], infection by adult *Fasciola* worms follows the ingestion of the encysted parasite *metacercariae* sitting on the leaves of a large variety of edible freshwater plants such as watercress, or other elements living in wet environments [75]. This makes *Fasciola* more frequent in sheep or cattle grazing on marshy or occasionally flooded pastures. Since *F. hepatica* requires a water temperature between 10°C and 30°C [71, 76] to complete its life cycle, the transmission of *fascioliasis* in the close environment of Els Trocs cave appears to be unlikely. Finally, flatworms of the genus *Paramphistomum* infect the stomachs of large or small ruminants [71]. Therefore, the eggs recovered in the sampled sediments could originate from the carcasses of domestic livestock or wild ruminants presumably eviscerated inside the cave. Before reaching its definitive host, these parasitic flukes also need a snail as an intermediate host and have a free-living external phase in water and plants. Hence, it is more plausible that the parasitic infection originally occurred in the lowlands. In sum, the ecological data derive from the Trematodes point towards the occupation of different ecosystems, in agreement with the practice of mobile herding. Also, the exponential increase in the number of collected eggs per sample and in biodiversity along the chronological sequence may be reflecting an intensifying frequency of people and animals using the cave that might have potentially favoured the outburst and development of new pathogens and contagious stages over time [77, 78].

## Discussion

The analytical proxies followed in this paper have focused on providing a solid corpus of data to investigate the caprine farming and herd management strategies developed by the human groups that occupied the Els Trocs cave throughout the Neolithic. By discussing the results from a comprehensive and diachronic perspective we believe we can gain a better understanding of animal husbandry practices involving the use of the Pyrenean uplands and their dynamics and in turn, improve our knowledge on the Neolithisation process of this ecosystem.

### Trocs I

The sheep:goat ratio highlights the absolute dominance of sheep in the Early Neolithic faunal assemblage. The estimated harvest profiles document perinatal losses and what appears to be the preferential slaughtering of immature and sub-adult caprines, presumably males but perhaps also females rejected for breeding. Based on the dental data (Table 3), individuals aged less than 6 months (classes A and B) represent 37% of the dead animals while the peak of mortality corresponds to sheep and goats between 6–12 months of age (class C: 24%). The results of the epiphyseal fusion (Table 2) indicate a larger mortality rate among the youngest cohort (group A- under 6 months: 49%) probably due to the counting of perinatal remains that are underrepresented in the dental data. Hence, according to one or other methodology the survival rate of caprines after their first year of age would range from 31 to 39%. The lack of great discrepancies between bone epiphyseal and dental data attests to limited recovery and preservation biases. In addition, the scarcity of animals of 3 years of age and older (Payne's classes E, F, G and H; 18% in total) would indicate that young adults, near their optimum meat weight, and adults, reaching the end of their reproductive life, were killed more frequently somewhere else at other time of the year. In this respect, the isotopic results demonstrate that sheep

practised vertical mobility in the Central Pyrenees since the late sixth millennium cal. BC. All three sampled specimens from Trocs I were certainly involved in this husbandry regime after their first year of age. Since the mandibular wear stage of ET Ovis 01 shows an age-at-death between 12–18 months, it may be assumed that this young lamb did not return to the lowlands after its first journey. On the contrary, ET Ovis 06 (2–3 years) and most probably, ET Ovis 02 (3–4 years) recurrently practiced vertical mobility over their lives. Thus, the mortality profiles conform with a 'truncated' pattern, reflecting the seasonal management strategy of the mobile herders. Whether it may also indicate the specialized use of Els Trocs cave for lambing and dairying production as it has been suggested for Neolithic caves of southern France, on the other side of the Pyrenees [62, 79, 80], requires a wider explanatory framework. For that reason, bringing into the discussion the ethnographic work on traditional transhumance in the Spanish Pyrenees may be elucidating [81–83].

These documentary sources point to late May and early June as the usual months when sheep flocks from the pre-Pyrenean territories began the journey to the uplands. By this time of the year, the harsh winter weather conditions of the mountain ecosystem (*i.e.*, snow and below 0˚ C temperatures) had diminished considerably, granting access to 'intermediate' pastures located between 1400 m and 1700 m a.s.l., where the shepherds and their livestock remained until the middle of July or even early August, before departing to higher grazing areas above 2000 m a.s.l. Likewise, such 'intermediate' pastures could be used for several weeks in October on the way back to the lowlands. Recent palaeoenvironmental studies on the southern side of the Axial Pyrenees highlight the fact that winter temperatures throughout the Holocene were much more stable than the summers, which between the interval 5500–3200 cal. BC hold the warmest temperatures associated with the Holocene climatic Optimum [84, 85]. Therefore, it may be argued that winter climatic constraints in the early Neolithic would not have differed greatly from those in the 19th-20th centuries. However, it is recognized that the historical large summer grasslands occupying the subalpine and montane belts of the Pyrenean mountain range are the result of progressive human-induced deforestation activities [86]. On this matter, geological surveys performed outside the Els Trocs cave have identified different episodes of vegetation burning dated to the late sixth millennium cal. BC [87], that would be compatible with small-scale anthropic deforestation events, potentially aiming at the procurement of pastures nearby. Together with few other cases [88, 89], these may represent the oldest anthropic alterations of the Pyrenean subalpine landscape, preceding in 3500 years the true and permanent effects of human societies on this ecosystem, which from 2000 cal. BC onwards are clearly recorded both in peat bogs [85, 90] and archaeological sites [91]. Admittedly, early forest clearings would have had a limited effect and occurred in reduced areas, but together with the archaeological evidences found at high altitude [22] they shed light on the complementarity of mountain grazing management and livestock cycles that would have been key factors in the Neolithisation process of the Pyrenean uplands.

Human intervention on this natural ecosystem suggests a growth in the size of the herds as well as the existence of social mechanisms for the coordination of separate activities within the early Neolithic small-scale farming communities. Altitudinal husbandry practices would have potentially shaped flexible and periodic interactions between householders from the same or different settlements for whom the use of seasonal highland pastures would have represented an opportunity to both overcome the difficulties of keeping viable breeding herds and share herding labour. In this context, it is plausible to hypothesize a division of tasks through the specialization of certain individuals or groups. The location of Els Trocs cave would be ideal to explore the transitional grazing areas near the flatlands of the Selvaplana before departing to the highest grasslands in summer or on the way back to the lowlands. Also, the proximity of the salt-rich water sources of La Muria and Fonsalada de Espés would have provided a vital

resource for the herds. The limited size of the cavity and the cold and wet environment inside conform best with its use as a shelter during short episodes rather than it being a sheepfold primarily dedicated to lambing. The low incidence of helminthiases noted in Trocs I would sustain the sporadic and relatively scarce number of animals and people occupying the cave, too.

The ethnographic studies similarly underline that thanks to a scheduled late autumn birthing season few pregnant and lactating females integrated the transhumant flocks. The results of the isotopic analyses document that two of the three sampled sheep were born in early spring (ET Ovis 01 and 02) and another (ET Ovis 06), although its $\delta^{18}O$ sequence could not be modelled, would most probably belong to a late spring birth. Indeed, the occurrence of caprines culled in age classes A, B and C enables to infer a long-lasting lambing season with rams mating ewes probably at all stages of the oestrous cycle. In consequence, perinatal remains in Trocs I could be deemed as the accumulation of occasional failures over time, reflecting the effects of seasonal mobility and the adaptation problems of the most vulnerable cohorts in the flocks (*i.e.*, pregnant females and pre-weaned lambs) to the mountain ecosystem. Neonatal lambs are very sensitive to temperature and humidity because of their small volume, the absence of the insulating layer of lanolin and the immaturity of the thermal regulation mechanism [92, 93]. Notwithstanding, taking into account that the faunal assemblage recovered from this occupation phase results from a number of uncertain events occurring over a long period of time, including a massacre [31], we cannot exclude the deliberate slaughtering of some very young lambs as part of special practices [94], eventually related to that violent episode.

Finally, a specialization in dairying production seems unlikely. The mortality data do not agree with any of the two theoretical models proposed by Helmer & Vigne [62]. In their Type A milk model the peak of mortality lies within the animals of less than 2 months of age. Type B milk is mostly characterised by the presence of a few B lambs (2 to 6 months), a high quantity of C lambs (killed for meat) and in general, an important part of E-F older females slaughtered when their lamb or milk production decreases. We do not reject though that some animals could have been milked to satisfy the domestic needs of the herders over the summer months, in particular as lactating females would have been available. Residue analyses of potsherds, currently under study, will help to clarify this issue.

**Trocs II.** The Middle Neolithic occupation at Els Trocs cave shows continuity in the dominance of caprines, among which the relative frequency of goats increases. Subtle changes in the mortality profiles and the lambing season see new dynamics to the herding management strategies at this phase that, together with other forms of helminthiases, seem to suggest a growing reliance on husbandry activities in the local/regional farming economy.

The abundance of perinatal remains decreases to 44% (S4 Table) and the age-at-death distributions point to the improved survival of caprines beyond the first year of age, accounting to 53% and 46%, based on epiphyseal fusion and dental data, respectively. The dental data document also a lower frequency of A lambs (0–2 months), an older peak of mortality (stage D 1–2 years) and more slaughtered adult specimens (stage G 4–6 years; Table 3) than in Trocs I. The decline in the loss of the youngest cohorts may be signalling at the gradual adaptation of husbandry systems to the mountain environmental setting described above. In this vein, the isotopic data on mobility practices and lambing season are clarifying. On the one hand, they note that the three sheep specimens that could be modelled did not practice vertical mobility until their second year of age. ET Ovis 07 went to the uplands for the first time at about 16 months of age whereas ET Ovis 03 and ET Ovis 05 only participated in vertical movements in their second summer of life. On the other hand, the isotopic results evidence a more irregular sheep breeding pattern than in Trocs I. Data from ET Ovis 04 could not be modelled but considering that the position of the maximum $\delta^{18}O$ value occurred between 10 to 15mm in

distance from ERJ, this specimen could represent a late spring birth. ET Ovis 07 was born at the end of winter (probably in February) while ET Ovis 03 and 05 were born in middle autumn (probably in October; Fig 10). These autumn births follow the de-seasoning mating pattern recently detected in the Epicardial levels of the French sites of Grotte Gazel and Taï [95], revealing an extended breeding season for Neolithic sheep in North-eastern Iberia, too. In our case, management of this sheep reproductive regime would have promoted the greater availability of weaned and stronger lambs by the time of departure to the summer grazing areas. However, the isotopic data on mobility note a delayed incorporation of autumn and late winter lambs to the mobile flocks. Given that autumn lambs have a long prepubertal period, it is reasonable to envisage this delay as part of risk minimisation strategies aiming at improving the size and the productive capacity of the herds as well as a way to ensure the supply of seasonal animal resources (*i.e.*, milk and tender meat) over a longer period in the year in the permanent settlements [95]. The strategy of subdividing herds so that breeding females had access to the most nutritious pastures reinforces the pastoral specialization previously noticed in Trocs I and indirectly points to the large size of the herds. Although it is not possible to be categorical, the higher relative frequency of goats in Trocs II (S3 Table) may be hinting in this same direction as the ability of goats to lead sheep flocks is well recognised by traditional shepherds in the Mediterranean world [96]. The rising sum of helminth eggs, in particular of *Dicrocoelium* (that mainly infects sheep) and the appearance of new genera such as *Ascaris* and *Paramphistomum*, presumably signal at a more regular use of the cave by both people and livestock during this phase. Thus, our results seem to indicate that the number of farming communities integrated into the seasonal vertical husbandry system in the Central Pyrenees probably increased in the course of the fifth millennium cal. BC.

The proliferation of contemporary upland sites (*i.e.*, Coro Trasito [22], Cova Colomera [34], Cova Gran de Santa Linya [97], Cova del Parco y Margineda [98]) where the identification of archaeological dung-rich deposits has established their function as penning spaces would also come to sustain this working hypothesis.

In sum, there are grounds to suggest that during the fifth millennium cal. BC the exploitation of the upland grazing areas by specialised pastoral groups would have facilitated the maintenance of large caprine flocks. The new herding management techniques observed in Trocs II show the adaptation to the environmental constraints of the mountain ecosystem, but they also enable to infer the gradual anthropisation of the natural landscape. While natural passages would have been recurrently used over the seasonal journeys to connect the uplands and the permanent settlements, rock-shelters and caves such as Els Trocs were more regularly occupied by herders and their livestock than in the early Neolithic. Even if the limited environmental impact of these activities implies their relatively small-scale, the improvements on mobile caprine herding strategies signal at the design of a new territorial organization in the region after the successful spread of the Neolithisation process into high-mountain locations.

**Trocs III.** The faunal record from the last Neolithic occupation at Els Trocs cave reveals that although the pastoral economy was still primarily focused on sheep and goat herding cattle exploitation might have increased in the pre-Pyrenean territories. Caprine perinatal remains testify to the on-going arrival of pregnant females to the uplands but the harvest profiles show the lowest frequency of individuals of less than 6 months of age along the Neolithic sequence. Contrary to the pattern observed in the other phases, the results from the epiphyseal fusion and the dental data, as regards the mortality of this young age group, differ greatly (31%-6%, respectively; Tables 2 and 3), pointing to potential taphonomical biases that might have negatively affected the survival and recovery of caprine milk teeth in this faunal assemblage. Indeed, counts of loose and *in situ* lower milk teeth reveal that Trocs III holds the largest frequency of isolated caprine deciduous teeth among all phases (Trocs III 63%, Trocs II 36%

and Trocs I 27%). Such circumstance has probably hampered the dental sample available to estimate the age-at-death, whilst the postcranial elements of young lambs and kids, complete or in halves, were more easily recovered and recorded. Hence, the fall in the rate of mortality among the youngest cohort is not as sharp as the dental data suggest. The peak of mortality seen in Trocs II (animals in their second year of age) is now reinforced. Also, for the first time, caprines older than 3 years of age, presumably barren or old breeding females, that in the previous periods were preferentially slaughtered in the permanent settlements, account for more than a quarter of the assemblage (Table 3).

Isotope measurements in sheep teeth keep on showing a wide lambing season. ET Ovis 10 was born in the middle of autumn, whereas specimens ET Ovis 09 and Ovis 11 should represent births occurring in late winter and late spring. Perhaps, the most relevant piece of information obtained from this analytical proxy is that we have been able to document the participation of a 9-month lamb in vertical movements. ET Ovis 10 shows a clear altitudinal pattern registered since the beginning of the record on its M2. As this specimen was probably born in October, it surely departed to the uplands in its first summer of life. This result illustrates how management of sheep breeding cycles would represent a good technical adaptation of mobile husbandry regimes to the mountain ecosystem [99].

The trends observed towards the on-site slaughtering of older caprines and the increase in cattle frequency in relation to Trocs II most likely reflect a shift in the pastoral production strategies of the lowland mixed farming communities during the fourth millennium cal. BC. The exploitation of secondary products provided by sheep, goats and cattle might have been emphasized, along with their primary products. Arguably, these circumstances attest to widened mobile specialised husbandry systems, too. Cattle and caprines would have been managed separately due to their specific grazing and herding needs. This scenario envisages new advances in the anthropisation process of the Pyrenean mountain range, because diversified production goals and the plausible presence of shepherds and cowherds groups may have prompted an intensified use of the uplands, where finding feeding resources to ensure the good nutritional state of growing livestock numbers would have potentially been easier than in the lowlands. The statistical higher density of *Capillaria* sp. and *Dicrocoelium* sp. eggs, and in general, larger concentration and variety of helminths recovered from this occupation phase at Els Trocs cave most likely mirror such intensification.

## Caprine mobile herding and management: Implications for the Neolithisation process of the Southern Central Pyrenees

Archaeological work in the last decade has broadened our understanding of the spread of Neolithic way of life in the uplands of the Southern Central Pyrenees [22, 23, 25, 33, 88, 89, 98, 100, 101]. Traditionally, such high altitude mountain areas were mainly considered as transit zones [102, 103]. However, recent fieldwork has demonstrated that archaeological sites located in this ecosystem have a lot to contribute to the present debate on the Neolithisation process and the definition of early Neolithic communities [12, 13, 25, 30]. The reasons for that are twofold. Firstly, they are all novel sites that respond to the demands of the new agropastoral subsistence system and of the firmly established productive lifestyle. And, secondly, the dominance of domestic animal remains, mainly caprines, among the recovered faunal assemblages, highlights their strong relationship with the development of complex herding practices involving the seasonal exploitation of subalpine grazing resources. Hence, vertical mobility practices are a key issue in the interpretation of subsistence economic models entailing complementarity between the Pre-Pyrenean territories and the highlands.

Most studies dedicated to the Neolithisation process of the subalpine Pyrenean areas offer a rather general appraisal of herding practices due to the recovery of poor faunal assemblages. Attempting to investigate animal husbandry on the basis of only a single criterion such as the identification of a few remains of domesticated species on a site can be misleading. Therefore, our work, integrating the analyses of caprine culling profiles, sheep isotopic measurements and helminthiases at Els Trocs cave within the archaeological and local ethnographic frameworks, has provided unique datasets to aptly gain insights into the dynamics of mobile herding management, production strategies and the scale of husbandry activities in this high-mountain area from the late sixth millennium cal. BC to the end of the fourth millennium cal. BC and consequently, better understand the anthropisation process of the upland territories.

The faunal spectrum from the three occupation phases is dominated by the main domesticate taxa–sheep, cattle and goats. Pig raising is not demonstrated since the morphological and size data retrieved from the remains recorded as suids (currently under study) point to wild boar as the only species represented. With the low contribution of other game and wild taxa hunting would have played a marginal role. Sheep always outnumber goats, but the latter increase from the Middle Neolithic onwards. The low management costs of caprines and their ease of mobility must have induced the human groups visiting the cave to primary focus their pastoral activities on these species.

The culling profiles and the isotopic data converge to evidence diverse caprine-rearing systems (*i.e.*, sedentary and mobile) as well as the implementation of herd management strategies aiming to gradually improve the adaptation of mobile herding to the mountain ecosystem. Thus, it may be argued that managing different sheep reproductive cycles as suggested by the record of autumn births from Trocs II onwards could have been one of the factors behind the progressive decline in perinatal mortality and the increased survival rate of caprines older than six months of age. The promotion of autumn births probably contributed to reducing the risks of losses as more weaned lambs and post-lactating females would be ready to initiate the journey in late spring. Also, bearing in mind that sheep and goats are milk producers, the presence of lactating females in the lowland permanent settlements, during the winter and early spring, fits well with an animal farming economy with more diversified production goals than the meat-oriented age-at-death patterns at the cave reflect. This in turn, gives support to the idea of a subsistence productive model in which pastoral resources were more intensively managed through time. The results of the palaeoparasitological analyses would warrant this hypothesis, too. The increase in the number of helminth eggs per sediment sample and in the taxonomical variety along the occupation sequence were interpreted in terms of the progressive proliferation of people and livestock (mainly sheep) seasonally exploring the subalpine grazing areas.

Based on the Neolithic herd management strategies for sheep, goat and cattle in southern France, Helmer et al. [79] and Bréhard et al. [80] argue that the functional complementarity between cave and open-air sites hints at the existence of mobile and sedentary regimes geared towards production exchange as well as to individuals or small groups specialized in herding activities since the middle of the fifth millennium cal. BC. Similarly, the fast dispersion of the Neolithic way of life after the arrival of the first farming communities in the Iberian Peninsula testifies to interregional contacts and communication networks [12, 104], that may have encompassed multidirectional and variable distance journeys. No matter how settled a community may have been, there would have always existed segments that had greater mobility [105]. The proxies analysed in this paper have revealed the seasonal and recurrent arrival of shepherds and their domestic livestock to the subalpine-alpine areas of the Southern Central Pyrenees from the early Neolithic. Indeed, the fact that we have been able to detect the practice of sheep vertical mobility at annual intervals in different animals, evidences this was not an occasional husbandry strategy but that it was an integral part of the farming economic model.

For this reason, we can affirm that mobile herding unquestionably contributed to the expansion of the new productive way of life throughout one of the most important mountain ecosystems in the Iberian Peninsula.

Vertical mobility would imply the periodic displacement of livestock and people along routes that linked the uplands with the farming communities located at lower altitudes. Control of these communication networks together with the management of the local natural resources (i.e., water, pastures) would have been major issues in securing the success of an economic activity that although it might have been practised quite randomly and at small-scale in the early Neolithic, according to our results, it was maintained and widened over time. Thus, the process of intensification detected in the archaeological record of Els Trocs cave has aided to visualize how seasonal altitudinal husbandry beyond the area of origin of the sedentary Neolithic communities played a role in the anthropisation of the subalpine and alpine belts of the central Pyrenees. Notwithstanding, this is just one piece of a complex puzzle that is still poorly understood, requiring extensive analyses of bioarchaeological samples and geoarchaeological studies of upland Neolithic substrates.

Obviously, the seasonal journeys during the Neolithic would not have encompassed the long-distance and large-scale movements of historical transhumance, but as suggested in this work they entailed social and economic interactions inside individual farming communities, and most probably among different communities. Labour costs associated with vertical husbandry strategies would be unsustainable for household farmers, most likely engaged in cultivation and perhaps owners of a small number of different livestock. From this perspective, we think that our results support the existence of specialised individuals or herder groups within the early mixed-farming communities settled in the lower-altitude pre-Pyrenean territories who were primarily interested in enhancing the feasibility of the local breeding herds of caprines.

This contribution, although based on the study of a single site, has enabled to draw attention to the complex nature of caprine mobile herding that not only motivated crucial modifications in the annual pastoral cycle but also required advanced management skills, control of the territory and the articulation with the other activities integrated in the farming systems of the lowlands. The progressive implementation of this husbandry system reveals the existence of a wide range of management strategies. Other multi-proxy analytical approaches at a regional scale would help to define them in more detail. Only then we would be able to have a fine-resolution picture of the spectrum of subsistence models that accompanied the anthropisation process of the Southern slopes of the Pyrenean mountain range.

## Supporting information

**S1 Fig. Intra-tooth variation of oxygen isotope values ($\delta^{18}O$) and carbon ($\delta^{13}C$) visible in sequentially sampled enamel bioapatite from two modern sheep performing vertical movements from the Ebro river valley to the Central Pyrenees [59].**
(TIF)

**S1 Table.** A) Radiocarbon dates from Els Trocs cave with lab references numbers (Mams = Mannheim AMS facility at the Curt-Engelhorn-Centre for Achaeometry; Beta = Beta Analytic), stratigraphic context, materials dated, and available isotopic data. Calibration is with OxCal v.4.4.2 (https://c14.arch.ox.ac.uk (1)) using IntCal20 atmospheric curve (2). B) Results of the Bayesian model (Phase sequential analysis) for the radiocarbon dates from Els Trocs cave. Calibration and modelled is with OxCal v.4.4.4 (https://c14.arch.ox.ac.uk (1)) using IntCal20 atmospheric curve (2).
(DOCX)

**S2 Table. Cova de Els Trocs.** Number of faunal remains (NR) in each Neolithic phase.
(DOCX)

**S3 Table. Cova de Els Trocs. Numbers and ratios of sheep (O) and goats (C) and percentages of sheep.** The numbers of deciduous lower fourth premolars (dLP4), scapulae (SC), distal humeri (dHU), proximal radii (pRA), distal metacarpals, (MTC), distal metatarsals (MTT), astragali (AST) and calcanea (CAL) are given as x: y, where x = number of sheep teeth or bones and y = number of goat teeth or bones.
(DOCX)

**S4 Table. Caprines MNE and percentages of perinatal elements in each Neolithic phase.**
(DOCX)

**S5 Table. Sheep specimens from Els Trocs cave included in the isotopic study.**
(XLS)

**S6 Table. Sediment samples from Els Trocs cave selected for the palaeoparasitological analyses.**
(XLSX)

**S7 Table. Results from the calculation of the best fit (by method of least squares) for combined variation of $X$ (period), $A$ (amplitude), $x_0$ (delay) and $M$ (mean) when the model is applied to $\delta^{18}$O M2 sequences obtained in sheep specimens from Els Trocs.** The Pearson's correlation coefficient ($R$) is also indicated.
(XLS)

**S8 Table. Results of the Fisher's exact test on the helminth eggs samples (all, *Ascaris* sp., *Capillaria* sp., *Dicrocoelium* sp., *Fasciola* sp., *Paramphistomum* sp., and *Trichuris* sp.), by occupation phase.**
(DOCX)

**S9 Table. Individual isotope results for each sampled specimen.**
(XLS)

**S1 File. Material and methods.** Supplementary material.
(DOC)

**S2 File. Main results from our pilot study on sequential sampling molar teeth for isotopic analysis of modern transhumant sheep in Iberia.** Supplementary material.
(DOC)

## Acknowledgments

Technical assistance by D. Fiorillo and scientific supervision from Dr M. Balasse (both UMR7209/MNHN) with the stable isotope analyses performed on sheep teeth samples carried out at the Service de Spectrométrie de Masse Isotopique du Muséum national Histoire naturelle of Paris, France (SSMIM) are gratefully acknowledged. We thank M.T. Sebastià and F. de Bello for providing information on pastures vegetation. P. Martín-Gómez and the Instituto de Formación Agro-ambiental are acknowledged for sharing isotope values in precipitation for Jaca, and J. Rodríguez-Arévalo and M.F. Díaz-Teijeiro for providing isotope data for Zaragoza, collected by the REVIP and run by CEDEX in cooperation with the Spanish Meteorological Agency (AEMET). Beatriu de Pinós, Post-doctoral Fellowship (2016–00346) and Research

Group SGR-836 are equally acknowledged. We thank Elena López-Romero and Esther Checa for their help while recording the archaezoological data.

## Author Contributions

**Conceptualization:** Cristina Tejedor-Rodríguez, Marta Moreno-García, Manuel Rojo-Guerra.

**Data curation:** Marta Moreno-García, Carlos Tornero.

**Formal analysis:** Marta Moreno-García.

**Funding acquisition:** Marta Moreno-García, Leonor Peña-Chocarro, Kurt. W. Alt, Manuel Rojo-Guerra.

**Investigation:** Cristina Tejedor-Rodríguez, Marta Moreno-García, Carlos Tornero, Alizé Hoffmann.

**Project administration:** Manuel Rojo-Guerra.

**Resources:** Cristina Tejedor-Rodríguez, Íñigo García-Martínez de Lagrán, Héctor Arcusa-Magallón, Rafael Garrido-Pena, José Ignacio Royo-Guillén, Sonia Díaz-Navarro.

**Supervision:** Marta Moreno-García, Leonor Peña-Chocarro, Kurt. W. Alt, Manuel Rojo-Guerra.

**Visualization:** Cristina Tejedor-Rodríguez, Marta Moreno-García, Carlos Tornero, Alizé Hoffmann, Héctor Arcusa-Magallón.

**Writing – original draft:** Marta Moreno-García.

**Writing – review & editing:** Cristina Tejedor-Rodríguez, Marta Moreno-García, Carlos Tornero, Manuel Rojo-Guerra.

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
