## [Decision Letter · Decision Letter 0]

29 Sep 2020

PONE-D-20-25388

Investigating Neolithic caprine husbandry in the Central Pyrenees: insights from a multi-proxy study at Els Trocs cave (Bisaurri, Spain)

PLOS ONE

Dear Dr. Moreno-García,

Thank you for submitting your manuscript to PLOS ONE. After careful consideration, we feel that it has merit but does not fully meet PLOS ONE’s publication criteria as it currently stands. Therefore, we invite you to submit a revised version of the manuscript that addresses the points raised during the review process.

All comments need to be addressed before re-submission.

We look forward to receiving your revised manuscript.

Kind regards,

Peter F. Biehl, PhD

Academic Editor

PLOS ONE

Journal Requirements:

2. In your Methods section, please ensure you have provided information regarding the permits you obtained for site and collections work. Please ensure you have included the full name of the authority that approved the field site access and, if no permits were required, a brief statement explaining why.

Additional Editor Comments (if provided):

Your manuscript has now been seen by two referees, whose comments are appended below. You will see from these comments that while the referees find your work of potential interest, one reviewer raised substantial concerns that must be addressed. In light of these comments, we cannot accept the manuscript for publication, but would be interested in considering a revised version that addresses these serious concerns.

We hope you will find the referees' comments useful as you decide how to proceed. Should presentation of further data and analysis allow you to address these criticisms, we would be happy to look at a substantially revised manuscript. However, please bear in mind that we will be reluctant to approach the referees again in the absence of major revisions.

Reviewers' comments:

Reviewer's Responses to Questions

**Comments to the Author**

1. Is the manuscript technically sound, and do the data support the conclusions?

Reviewer #1: Yes

Reviewer #2: Yes

2. Has the statistical analysis been performed appropriately and rigorously? 

Reviewer #1: Yes

Reviewer #2: Yes

3. Have the authors made all data underlying the findings in their manuscript fully available?

Reviewer #1: Yes

Reviewer #2: No

4. Is the manuscript presented in an intelligible fashion and written in standard English?

Reviewer #1: Yes

Reviewer #2: Yes

5. Review Comments to the Author

Reviewer #1: This is a well-executed and interesting multidisciplinary study on the development and intensification of pastoral activities during the Iberian Neolithic. I only have a few suggestions to help clarify certain points in the data presentation and argument.

1. In the abstract (lines 48-51) and introduction (lines 104-108), the authors present broader questions about how the spread of pastoralism may have led to the settlement, control, and anthropization of the landscape. The authors should revisit these questions more pointedly in the discussion and conclusion sections- there is certainly enough data to provide a more detailed answer.

2. Similarly, in lines 78-79 the authors write that animal husbandry could have been practiced at either the household level or been more specialized. The authors imply the different levels of specialization that developed throughout the Neolithic, but should explicitly address the household vs. group scale of these economic activities and possible change over time in their discussion/conclusion.

3. I found the faunal tables difficult to read and did not understand what the two % columns for each level were meant to represent.

4. The authors might consider including a figure or table summarizing the chronology and/or stratigraphy of the cave so that readers can more easily follow along.

5. Lines 860-863: While interesting, the sudden introduction of lithic evidence at the end of the paper without much further discussion is a bit jarring. The authors should either elaborate on this point, perhaps earlier in their discussion, or omit it.

Minor Points

1. The authors should consider recalibrating their 14C dates with the newly published calibration curve.

2. Line 141: It’s unclear if the floor fashioned from potsherds is intentional or not.

3. Line 170: Could the authors please more clearly define “combustion structure”?

4. Line 524: What is the p-value, etc of the Fisher’s exact test?

5. Line 557: I assume the authors meant “this does not mean it could not have happened”?

6. Line 741: What statistical test was used to demonstrate significance?

7. Lines 783-784: What is the citation for this statement?

8. These are very minor grammatical/spelling errors I caught while reading.

line 69: “adaptive” instead of “adaptation”

line 87: “finds it hard challenging” rewrite for clarity

line 104: “fill” instead of “fulfill”

line 180: “cave” should be plural

line 194: insert “were” between “hearths” and “successively”

line 195: “was” instead of “were”

line 196: insert “an” between “Such” and “event”

line 404: delete “how”

line 822: insert “the” between “while” and “other”

Reviewer #2: I thank the authors for submitting and sharing new data that contributes to the understanding of neolithisation in the Pyrenees. Overall, I do not disagree with the conclusions drawn from the data (however, these need to be rephrased as suggestions). The paper presents original research and the data that has been submitted appears to be performed to a rigorous standard. However, in my opinion the results and discussion could be strengthened in a number of ways.

My comments to the authors are two-fold: larger clarifications regarding research design and analyses and specific corrections throughout the paper.

Two major technical issues arise throughout the paper. First, in all reporting of tooth position, standard literature dictates the number of the molar be placed in either subscript (mandibular) or superscript (maxillary) based on where the tooth originated (I apologize if this adjustment has already been made and the PDF submission altered that formating). Please correct throughout. Second, the commas reported in all numerical values needs to be changed to decimal periods (I.e., -10.7 instead of -10,7). Please correct throughout.

The opening argument for this research is strong. Some expansion on the arrival of domesticates could strengthen the importance of the site in question. While this is a scientific-based journal, I was surprised to see little literature review on carbon and oxygen isotopic analyses in application to archaeological material. This would strengthen the paper to allow more readers who are non-experts to have greater accessibility to engage with the data and conclusions.

Lines 190-191: In my opinion, additional citations are needed to support the larvae assertion in connection to seasonality.

Lines 245-246: This statement is asserting seasonal mobility, but the previous lead up in the research design suggests this is being tested.

Lines 249-253: I don’t see a need for this section, as this has already been clearly laid out for the reader.

In regard to the materials and methods section, I have two troubling issues. First, the eleven individuals sample size in small, particularly if there were approximately 1,068 caprine teeth recovered. I expected a more robust sample size for the questions being addressed through this research. If this was a decision made based on preservation or economics, please make a note. This seems more like a pilot study sample size. Second, it needs to be made clear earlier (perhaps in the suggested isotopic literature review) that bioapatite is being sampled. Here resides my second question: If mobility, in this case vertical mobility, was a central aim laid out in the introduction and bioapatite was chosen for study, why wasn’t strontium analysis built into the research design? In my opinion, the analysis of strontium would have gone hand-in-hand with the aims of this study. Again, I recognize this may have been considered and deemed unfeasible (perhaps due to lacking baseline data), but I suggest this omission be briefly presented in this section. On a smaller note, I was happy to see the methods described in detail among the supplementary material, though a quick note on the sampling method and instrumentation should be included in the main paper.

In regard to the results section, given the aim of the research, a distinction in the averages between the M2 and M3 samples is warranted. The associated graphs need to have a legend or key (I.e., distinguish carbon and oxygen as either an open circle or filled circle, as it was not clear which was which).

Line 414: ET Ovis 03 is included in the associated graph. This statement should include 03, since the statement covers the broad analysis of the M3’s.

Further distinction in the presented results between stratigraphic phases would strengthen the assertions made in the discussion section.

Line 494: I’m hesitant in these assertions based on the low sample number and would recommend the phrasing is changed to “suggests”.

Continuing in the results section, in my opinion, the results presentation of the isotopic results is lacking. I’m troubled that the results did not include a discussion on vegetation and water intake. The omission of results in connection with animal diet was surprising given the aims at reconstructing herd management practices. In the discussion section, a clear connection is being drawn between the parasitic data and vegetation and water consumption, which the presented carbon and oxygen isotopic data could speak to, but these results were not presented or discussed. Further, I believe this will alleviate, in my opinion, an issue on the connection between the isotopic study and the palaeoparasitological analysis and would overall strengthen the assertions between these two studies presented in the conclusions of the paper. I recommend the results are presented in correlation with vegetation intake (C3 versus C4) and water intake, which could further strengthen the argument for vertical mobility.

To address my above answer on data availability, the authors need to provide a supplementary table with all the individual isotope sample results for each specimen, not just the provided summary table. It is also highly recommended that the authors provide a supplementary table with full instrument analytical details for the isotopic analyses.

Line 721-723: I am unconvinced that your data supports this conclusion, primarily based on the large number of undetermined caprine remains in the faunal assemblage (which, while asserted as probable sheep, cannot be determined definitively either way and may be comprised of primarily goat remains). Also, the multitude of factors that can result in an increase in goats leads me to question sufficient data to back this assertion.

Overall, I urge the authors to take these comments into consideration and make those revisions prior to publication. Again, I support the new data presented and the majority of conclusions drawn and believe this research will provide a meaningful contribution to the scientific community.

6. PLOS authors have the option to publish the peer review history of their article (what does this mean?). If published, this will include your full peer review and any attached files.

Reviewer #1: No

Reviewer #2: No

---

## [Author Response · Author response to Decision Letter 0]

11 Nov 2020

REVIEWER 1 COMMENTS:

1. In the abstract (lines 48-51) and introduction (lines 104-108), the authors present broader questions about how the spread of pastoralism may have led to the settlement, control, and anthropization of the landscape. The authors should revisit these questions more pointedly in the discussion and conclusion sections- there is certainly enough data to provide a more detailed answer.

RESPONSE. We thank the reviewer for this comment. We have revisited this issue and new text has been included in the manuscript. See lines 806-817, 855-863 and 952-967.

2. Similarly, in lines 78-79 the authors write that animal husbandry could have been practiced at either the household level or been more specialized. The authors imply the different levels of specialization that developed throughout the Neolithic, but should explicitly address the household vs. group scale of these economic activities and possible change over time in their discussion/conclusion.

 RESPONSE. This issue has been addressed more specifically as suggested by the reviewer. Please, see the text in lines 702-710, 784-790, 806-808, 968-977.

3. I found the faunal tables difficult to read and did not understand what the two % columns for each level were meant to represent.

RESPONSE. A new footnote has been included in Table 1 to describe more precisely the two % columns. It reads as follows: b Relative frequency based on % NISP counts in identified taxa; c Relative frequency based on % NISP counts in husbanded taxa only.

4. The authors might consider including a figure or table summarizing the chronology and/or stratigraphy of the cave so that readers can more easily follow along.

RESPONSE. Following the suggestion made we have included (Lines 139-145) a new figure (Fig. 2) that summarizes the chronology/ stratigraphy of the cave after performing a Bayesian analysis by applying a “phase sequential” model to the radiocarbon dates presented in “S1 Table”. Dates have been grouped by chronological phases according to the stratigraphic unit they derive from. In that way, the three Neolithic occupation phases studied along the paper are clearly defined.

Inclusion of this new figure in the manuscript means that successive figures have been renumbered.

5. Lines 860-863: While interesting, the sudden introduction of lithic evidence at the end of the paper without much further discussion is a bit jarring. The authors should either elaborate on this point, perhaps earlier in their discussion, or omit it.

RESPONSE. This sentence regarding the lithic evidence has been removed from the manuscript.

Minor Points

1. The authors should consider recalibrating their 14C dates with the newly published calibration curve.

RESPONSE. 14C dates have been recalibrated with the newly calibration IntCAL20 (Reimer et al. 2020) and software OxCal v.4.4.2 (https://c14.arch.ox.ac.uk/oxcal/OxCal.html; Bronk Ramsey, 2020). Accordingly, S1 Table has been modified.

2. Line 141: It’s unclear if the floor fashioned from potsherds is intentional or not.

RESPONSE. This has now been clarified in Line 148. 

3. Line 170: Could the authors please more clearly define “combustion structure”?

RESPONSE. “Combustion structure” has now been substituted by “hearth” in Line 177.

4. Line 524: What is the p-value, etc of the Fisher’s exact test?

RESPONSE. The p-value of the Fisher’s exact test is specifically mentioned in S8 Table but we have included it now in the text (Line 579).

5. Line 557: I assume the authors meant “this does not mean it could not have happened”?

RESPONSE. The reviewer is correct. The mistake has now been corrected in Line 613.

6. Line 741: What statistical test was used to demonstrate significance?

RESPONSE. We meant that results differ greatly. The wording has been changed in Line 829.

7. Lines 783-784: What is the citation for this statement?

RESPONSE. Two new citations have been included in support of that statement. 

Baldellou V, Utrilla P. Arte rupestre y cultura material en Aragón: Presencias y ausencias, convergencias y divergencias. Bolskan, (Jornadas técnicas sobre arte rupestre y territorio arqueológico). 1999; 6: 21–37.

Utrilla P, Mazo C. La Peña de las Forcas (Graus, Huesca). Un asentamiento estratégico en la confluencia del Ésera y el Isábena. Zaragoza: Universidad de Zaragoza, Monografías Arqueológicas, Prehistoria, 2014; 46.

8. These are very minor grammatical/spelling errors I caught while reading.

line 69: “adaptive” instead of “adaptation”

line 87: “finds it hard challenging” rewrite for clarity

line 104: “fill” instead of “fulfill”

line 180: “cave” should be plural

line 194: insert “were” between “hearths” and “successively”

line 195: “was” instead of “were”

line 196: insert “an” between “Such” and “event”

line 404: delete “how”

line 822: insert “the” between “while” and “other”

RESPONSE. We thank the reviewer for his/her careful examination of the text. All these grammatical/spelling mistakes have been corrected and the expression in line 87 has been rewritten.

REVIEWER 2 COMMENTS:

My comments to the authors are two-fold: larger clarifications regarding research design and analyses and specific corrections throughout the paper.

Two major technical issues arise throughout the paper. First, in all reporting of tooth position, standard literature dictates the number of the molar be placed in either subscript (mandibular) or superscript (maxillary) based on where the tooth originated (I apologize if this adjustment has already been made and the PDF submission altered that formating). Please correct throughout. 

 RESPONSE. We thank the reviewer for his/her careful reading of the manuscript but the assertion as “major” of the two technical issues he/she points out is, in our opinion, unjustified and exaggerated. Regarding the use of a subscript or superscript number to refer to mandibular or maxillary teeth, we are aware that this is the standard practice to avoid confusion whenever both sets of teeth are analyzed. In the methodology section is clearly stated that only mandibular teeth have been used in our aging (line 279) and isotopic analyses (line 293). Therefore, the use of such numerical format seems somehow redundant. In addition, a quick revision of similar works published in PloS ONE (i.e., Gron KJ, Montgomery J, Rowley-Conwy P. Cattle Management for Dairying in Scandinavia’s Earliest Neolithic. PLoS ONE. 2015;10(7):e0131267; Vaiglova P, Halstead P, Pappa M, Triantaphyllou S, Valamoti SM, Evans J, et al. Of cattle and feasts: Multi-isotope investigation of animal husbandry and communal feasting at Neolithic Makriyalos, northern Greece. PLOS ONE. 2018;13(6):e0194474.) shows no subscript formatting is followed. For these reasons, the original manuscript has not been altered.

Second, the commas reported in all numerical values needs to be changed to decimal periods (I.e., -10.7 instead of -10,7). Please correct throughout.

RESPONSE. The decimal separator has been changed to a dot in all numerical values.

The opening argument for this research is strong. Some expansion on the arrival of domesticates could strengthen the importance of the site in question. 

 RESPONSE. We think that the positive assertion made by the reviewer on our opening argument lays in the clear contextualization of our research. Between lines 54 and 71 there are 28 citations of works that deal with the first Neolithic communities in Iberia and the expansion of the Neolithisation process. The reviewer is probably aware that while they thoroughly discuss the different Neolithisation models that have been proposed for this geographical area, references to the arrival and spread of domesticate taxa (mainly sheep) relate in most cases to the radiocarbon dates obtained from them. The management and exploitation of husbanded species have hardly been discussed in those models. A recently published paper that notes the limited knowledge on the early introduction of domesticates in Iberia has been added to the original manuscript (Saña et al. 2020 (5)). Also, as pointed by Martins et al 2015 (3), the taxonomic identification of the caprine samples that have provided the oldest dates needs to be revised. In our opinion, the significance of the archaeological site here studied is noted between lines 91 and 94, according to the arguments presented in the previous paragraph, and more extensively in the section ‘Site description, chronology and findings’. 

While this is a scientific-based journal, I was surprised to see little literature review on carbon and oxygen isotopic analyses in application to archaeological material. This would strengthen the paper to allow more readers who are non-experts to have greater accessibility to engage with the data and conclusions.

RESPONSE. We appreciate the comments made by reviewer 2. However, it must be understood that carbon and oxygen isotopic analyses are just one of the analytical proxies in this study. Making a literature review on the application of this methodological approach to archaeological material in general and not following the same strategy with the other methodological approaches would create an unjustified unbalance among the three proxies. Major reference works are cited in the main manuscript in relation to the research questions investigated in the paper. In particular, (56) Balasse et al., 2012 Archaeometry; (57) Balasse et al., 2003 JAS; (58) Tornero et al., 2013 JAS; (63) Balasse et al., 2009 Environmental Archaeology; (64) Tornero et al., 2016 JAS-Reports; (65) Blaise and Balasse, 2011 JAS; (95) Tornero et al., 2020 Scientific Reports. Overall, detailed methodological aspects (as the reviewer recognizes below) and related literature are thoroughly described in S1 File 'Materials and Methods' and in S2 File ‘Main results from our pilot study on sequential sampling molar teeth for isotopic analysis of modern transhumant sheep in Iberia’. We respect the opinion of the reviewer but we do not share his/her idea that a literature review on carbon and oxygen isotopic analyses would engage non-expert readers with our data and conclusions. Actually, the interest of the isotopic analyses results lay in their interpretation within the framework provided by the culling profiles, the archaeological data and the study carried out with modern transhumant sheep (S2 File). We think that readers will feel attracted to our work because on the one hand, our research aims, methods and results are honestly and clearly stated in the abstract of the manuscript and on the other hand, as the reviewer states at the end of his/her comments, ‘this research provides a meaningful contribution to the scientific community’. 

Lines 190-191: In my opinion, additional citations are needed to support the larvae assertion in connection to seasonality.

RESPONSE. Two new references have been added.

Harrington R, Stork NE, editors. Insects in a Changing Environment. 17th Symposium of the Royal Entomological Society. London: Academic Press; 1995.

Skartveit J. The larvae of European Bibioninae (Diptera, Bibionidae). Journal of Natural History. 2002;36(4):449-485.

Lines 245-246: This statement is asserting seasonal mobility, but the previous lead up in the research design suggests this is being tested.

 RESPONSE. In our opinion, the reviewer has misunderstood the statement in those lines. There, we make reference to the seasonal occupation of the cave by different human groups throughout the Neolithic sequence. Such fact has already been demonstrated in other publications (Lancelotti et al 2014) and it is not questioned or revisited here. It is also mentioned in the abstract (line 33) and explained in the ‘Site description, chronology and findings’ section (lines 166-167, 197-200). If the Academic editor considers that the word “seasonally” is confusing it can be removed from that sentence.

Lines 249-253: I don’t see a need for this section, as this has already been clearly laid out for the reader.

 RESPONSE. We agree that this section is redundant. We have thus removed it from the text.

In regard to the materials and methods section, I have two troubling issues. First, the eleven individuals sample size in small, particularly if there were approximately 1,068 caprine teeth recovered. I expected a more robust sample size for the questions being addressed through this research. If this was a decision made based on preservation or economics, please make a note. This seems more like a pilot study sample size.

 RESPONSE. We must confess that the comment on the small sample size used in the isotopic analyses in relation to the total number of 1068 teeth puzzles us for a number of reasons. Firstly, it cannot be expected that all the caprine teeth recovered from any archaeological site would be suitable for isotopic analyses. Given the number of juvenile animals in our assemblages, it is understood that deciduous teeth comprise a large part of the dental samples. Secondly, as it is clearly stated in the manuscript, we were exclusively interested in sheep because this is the dominant taxa in the archaeological assemblages. Therefore, the standard diagnostic morphological criteria were followed to make sure we were analyzing sheep samples (lines 296-298). Since it is not possible to discriminate sheep from goats using upper teeth, these dental remains were not considered. Thirdly, according to the isotopic literature cited (lines 304-310) and criteria explained in S1 File, unworn M2 and M3 are the optimal specimens for applying sequential series of isotopic analyses because their long crowns allow to obtain the desirable time period of analysis. Consequently, teeth placed in situ (that is, complete mandibles) rather than isolated teeth were selected. A quick look at Table 3 enables to see that 24%, 21% and 25% of the caprine mandibles in Trocs I, II and III, respectively belong to individuals in the second half of their first year of age. This is the age range when M2 is formed and erupting. In sum, all these criteria on sampling selection reduced the number of optimal specimens to eleven. Notwithstanding, following the recommendation made by the reviewer a note has been included at the beginning of the isotopic analysis section in S1 File to further clarify this issue.

Second, it needs to be made clear earlier (perhaps in the suggested isotopic literature review) that bioapatite is being sampled. 

 RESPONSE. Although the legend of Table 4 clearly states that bioapatite was being sampled, this issue has been clarified in the main text (lines 294-296).

Here resides my second question: If mobility, in this case vertical mobility, was a central aim laid out in the introduction and bioapatite was chosen for study, why wasn’t strontium analysis built into the research design? In my opinion, the analysis of strontium would have gone hand-in-hand with the aims of this study. Again, I recognize this may have been considered and deemed unfeasible (perhaps due to lacking baseline data), but I suggest this omission be briefly presented in this section. 

 RESPONSE. We acknowledge that strontium measurements are indeed an excellent analytical proxy in archaeology to investigate residential patterns in humans and, by extension, in animal populations. However, as the reviewer mentions, strontium analysis requires an important corpus of baseline data about the area of study, built from modern plant or animal samples that unfortunately, at present it is not available for this area of the Pyrenees. Also, the reviewer would agree with us that the potential success of strontium analyses to achieve our scientific objectives would depend on important isotopic differences between the upland and lowland regions. Such an issue has not been investigated for the time being but it would be worth considering in future studies once the provenience locations in the lowlands of the Neolithic sheep visiting Els Trocs cave are surely known. Accordingly, justifying why strontium analysis was not one of the analytical proxies in our study would potentially add useless background noise to our research design. We believe that our aims were fully and noticeably achieved by using δ13C and δ18O isotopic ratios. 

On a smaller note, I was happy to see the methods described in detail among the supplementary material, though a quick note on the sampling method and instrumentation should be included in the main paper.

 RESPONSE. The quick note on the sampling method and instrumentation has been added in the main paper (lines 312-318). 

In regard to the results section, given the aim of the research, a distinction in the averages between the M2 and M3 samples is warranted. The associated graphs need to have a legend or key (I.e., distinguish carbon and oxygen as either an open circle or filled circle, as it was not clear which was which).

RESPONSE. The suggested correction has been made (lines 407-412) and the legends in Figures 7 (previously 6) and 8 (previously 7) have been modified. They read as follows:

Fig 7. Results from the sequential analysis of oxygen (δ18OV-PDB, white circles) and carbon (δ13CV-PDB, black circles) isotope composition in sampled enamel bioapatite from Els Trocs cave sheep lower second molars (M2).

Fig 8. Sequential analysis of oxygen (δ18OV-PDB, white circles) and carbon (δ13CV-PDB, black circles) isotope composition in lower second (M2) and third (M3) molars in five of the sheep samples from Els Trocs cave.

Line 414: ET Ovis 03 is included in the associated graph. This statement should include 03, since the statement covers the broad analysis of the M3’s.

 RESPONSE. ET Ovis 03 is included in the associated graph but the statement in line 414 (now line 460) does not allude to the broad analysis of the M3s but only to the cases in which the inverse relationship between carbon and oxygen in M2s was not clear. ET Ovis 03 is mentioned in the next sentence because its M3 was analyzed to check if the non-vertical history detected in its M2 changed. It represents a different circumstance that is clearly explained in lines 456-457, 461-463. 

Further distinction in the presented results between stratigraphic phases would strengthen the assertions made in the discussion section.

 RESPONSE. A new paragraph has been added in the Results section following the recommendation of the reviewer (lines 495-503).

Line 494: I’m hesitant in these assertions based on the low sample number and would recommend the phrasing is changed to “suggests”.

 RESPONSE. The change has been made as suggested.

Continuing in the results section, in my opinion, the results presentation of the isotopic results is lacking. I’m troubled that the results did not include a discussion on vegetation and water intake. The omission of results in connection with animal diet was surprising given the aims at reconstructing herd management practices. In the discussion section, a clear connection is being drawn between the parasitic data and vegetation and water consumption, which the presented carbon and oxygen isotopic data could speak to, but these results were not presented or discussed. Further, I believe this will alleviate, in my opinion, an issue on the connection between the isotopic study and the palaeoparasitological analysis and would overall strengthen the assertions between these two studies presented in the conclusions of the paper. I recommend the results are presented in correlation with vegetation intake (C3 versus C4) and water intake, which could further strengthen the argument for vertical mobility.

 RESPONSE. With all due respect, we find that the assertion regarding the presentation of the isotopic results is totally unjustified. All the relevant and key information about the isotopic results are well-presented in the Results section, Tables and Figures. We admit though that since S2 File presents a detailed summary on how vegetation and water intake is affecting serial sequences these issues were not discussed in the main text. But, following the recommendation of the reviewer an explanation on diet has been added to correct this omission (lines 429-444).

To address my above answer on data availability, the authors need to provide a supplementary table with all the individual isotope sample results for each specimen, not just the provided summary table. It is also highly recommended that the authors provide a supplementary table with full instrument analytical details for the isotopic analyses.

 RESPONSE. To clarify this issue, we would like to state that the presentation of the isotopic data and results follows the publication standards used in similar studies. Please, see Guiry et al., 2016 JAS; Pilaar Birch et al., 2016 PLOS ONE; Tornero et al., 2016 Journal of Human Evolution; Makarewicz et al., 2017 JAS; Tornero et al., 2018 Quaternary International; Balasse et al., 2020 JAS; Tornero et al., 2020 Scientific Reports. Further to the maximum, minimum, mean and estimated range isotopic values presented in Table 4, results from all sampled molars and investigated sheep are plotted in Figures 7 (previously 6) and 8 (previously 7) in the main manuscript. However, following the requirement from the reviewer, we have added a table with the same results plotted in Figures 7 and 8 (S9_Table), and let its final publication open to the Academic Editor. Reference to S9 Table has been included in line 406.

As regards the publication of a supplementary table with full instrument analytical details for the isotopic analyses, we are absolutely astonished because we have never been asked to provide such details in any of our previous publications dealing with the isotopic analyses of archaeological bioapatite samples. S1 File explains that bioapatite samples were measured using a Keil IV device interfaced to a Delta V Advantage isotope ratio mass spectrometer (IRMS) at the Service de Spectrométrie de Masse isotopique du Muséum national d'Histoire naturelle (SSMIM) in Paris (France). Accuracy and precision of the measurements were checked using an internal laboratory calcium carbonate standard (Marbre LM normalized to NBS 19) and we obtained a high mean analytical precision of the Marbre LM, +0.02 ± 0.008 for δ13C and +0.04 ± 0.009 for δ18O values. The reviewer should know that these data validate our measurements. 

Line 721-723: I am unconvinced that your data supports this conclusion, primarily based on the large number of undetermined caprine remains in the faunal assemblage (which, while asserted as probable sheep, cannot be determined definitively either way and may be comprised of primarily goat remains). Also, the multitude of factors that can result in an increase in goats leads me to question sufficient data to back this assertion.

 RESPONSE. We agree that there is a large number of undetermined caprine remains in our faunal assemblage, but the data on the sheep:goat ratios based on the minimum number of elements that could be specifically identified according to well-described morphological and metrical criteria attests to the clear dominance of sheep over goats (lines 253-259, S3 Table). In archaeozoological studies, calculation of this ratio is a common and simple tool that allows to estimate the relative frequency of both species in any assemblage. In turn, this parameter is usually used to characterize the undetermined caprine fraction which in our case is mainly comprised by neonate and juvenile individuals. The working premise is that if among the adult population sheep dominate, it is likely that the juvenile animals represented in the assemblage would belong to this species. If few adult goats have been identified, it would be rather weird to assume a substantial presence of their offspring on site. For these reasons, we firmly believe that any archaeozoologist would find that the noted increase in the relative frequency of goats in the flocks during Trocs II and Trocs III is totally backed-up by the data provided. 

We certainly acknowledge that multiple factors may have influenced the herders’ decision to increase the number of goats. This hypothesis has been rephrased and a citation has been added (lines 787-790).

Overall, I urge the authors to take these comments into consideration and make those revisions prior to publication. Again, I support the new data presented and the majority of conclusions drawn and believe this research will provide a meaningful contribution to the scientific community.

---

## [Decision Letter · Decision Letter 1]

4 Dec 2020

Investigating Neolithic caprine husbandry in the Central Pyrenees: insights from a multi-proxy study at Els Trocs cave (Bisaurri, Spain)

PONE-D-20-25388R1

Dear Dr. Moreno-García,

We’re pleased to inform you that your manuscript has been judged scientifically suitable for publication and will be formally accepted for publication once it meets all outstanding technical requirements.

Kind regards,

Peter F. Biehl, PhD

Academic Editor

PLOS ONE

Additional Editor Comments (optional):

Reviewers' comments:

Reviewer's Responses to Questions

**Comments to the Author**

1. If the authors have adequately addressed your comments raised in a previous round of review and you feel that this manuscript is now acceptable for publication, you may indicate that here to bypass the “Comments to the Author” section, enter your conflict of interest statement in the “Confidential to Editor” section, and submit your "Accept" recommendation.

Reviewer #1: All comments have been addressed

Reviewer #2: All comments have been addressed

2. Is the manuscript technically sound, and do the data support the conclusions?

Reviewer #1: Yes

Reviewer #2: Yes

3. Has the statistical analysis been performed appropriately and rigorously? 

Reviewer #1: Yes

Reviewer #2: Yes

4. Have the authors made all data underlying the findings in their manuscript fully available?

Reviewer #1: Yes

Reviewer #2: Yes

5. Is the manuscript presented in an intelligible fashion and written in standard English?

Reviewer #1: Yes

Reviewer #2: Yes

6. Review Comments to the Author

Reviewer #1: (No Response)

Reviewer #2: This is a well-written and engaging manuscript. I appreciate the changes the authors made, and respect those that were challenged with supporting literature. The manuscript is strong and impactful, well done!

7. PLOS authors have the option to publish the peer review history of their article (what does this mean?). If published, this will include your full peer review and any attached files.

Reviewer #1: No

Reviewer #2: No

---

## [Editor Report · Acceptance letter]

9 Dec 2020

PONE-D-20-25388R1 

Investigating Neolithic caprine husbandry in the Central Pyrenees: insights from a multi-proxy study at Els Trocs cave (Bisaurri, Spain) 

Dear Dr. Moreno-García:

I'm pleased to inform you that your manuscript has been deemed suitable for publication in PLOS ONE. Congratulations! Your manuscript is now with our production department. 

Kind regards, 

on behalf of

Dr. Peter F. Biehl 

Academic Editor

PLOS ONE